# Synergistic Benefits of Joint Molecule Generation and Property Prediction

**Adam Izdebski**                                    *adam.izdebski@helmholtz-munich.de*
*Institute of AI for Health, Helmholtz Zentrum Munchen*
*Technical University of Munich, TUM School of Computation, Information and Technology*
*Faculty of Mathematics, Informatics and Mechanics, University of Warsaw*

**Jan Olszewski**
*Faculty of Mathematics, Informatics and Mechanics, University of Warsaw*

**Pankhil Gawade**
*Institute of AI for Health, Helmholtz Zentrum Munchen*

**Krzysztof Koras**
*Ardigen SA*

**Serra Korkmaz**
*Institute of AI for Health, Helmholtz Zentrum Munchen*

**Valentin Rauscher**
*Technical University of Munich*

**Jakub M. Tomczak**
*Eindhoven University of Technology*

**Ewa Szczurek**                                    *ewa.szczurek@helmholtz-munich.de*
*Institute of AI for Health, Helmholtz Zentrum Munchen*
*Faculty of Mathematics, Informatics and Mechanics, University of Warsaw*

**Reviewed on OpenReview:** *https://openreview.net/forum?id=jnzCOLyGOA*

## Abstract

Modeling the joint distribution of data samples and their properties allows to construct a single model for both data generation and property prediction, with synergistic benefits reaching beyond purely generative or predictive models. However, training joint models presents daunting architectural and optimization challenges. Here, we propose HYFORMER, a transformer-based joint model that successfully blends the generative and predictive functionalities, using an alternating attention mechanism and a joint pre-training scheme. We show that HYFORMER is simultaneously optimized for molecule generation and property prediction, while exhibiting synergistic benefits in conditional sampling, out-of-distribution property prediction and representation learning. Finally, we demonstrate the benefits of joint learning in a drug design use case of discovering novel antimicrobial peptides.

---

Code available at: https://github.com/szczurek-lab/hyformer

# 1 Introduction

Developing models that simultaneously excel in both generative and predictive tasks is a long-standing challenge in machine learning (Bishop, 1994; Jaakkola & Haussler, 1998; Lasserre et al., 2006). Joint models, which unify these tasks, offer synergistic benefits, including improved control over the generative process of the model, improved predictive robustness towards unseen, e.g., newly generated or out-of-distribution (OOD) data, and learning representations predictive of high-level molecular features (Nalisnick et al., 2019; Grathwohl et al., 2020; Cao & Zhang, 2022; Tomczak, 2022). These benefits are crucial for applications such as drug design, where success depends on balancing the generation of novel molecules from unexplored regions of the chemical space coupled with robust property prediction extrapolating towards the newly generated molecules (Grisoni, 2023; Steshin, 2023; van Tilborg et al., 2025).

However, molecule generation and property prediction are predominantly approached in separation. This division persists even though transformer-based models are state-of-the-art across both tasks (Bagal et al., 2022; Gao et al., 2024b; Irwin et al., 2022; Xia et al., 2023; Zhou et al., 2023). A likely reason is that joint training poses daunting challenges, as combining a generative and a predictive part into a single model may over-regularize both parts (Lasserre et al., 2006) or cause gradient interference between the generative and predictive objectives (Nalisnick et al., 2019). As a result, molecular models continue to forgo the potential benefits of joint learning. This raises a natural question, whether one can *develop a transformer-based joint model optimized for both generative and predictive performance, at the same time offering the synergistic benefits of joint learning?*

To address this challenge, we introduce HYFORMER, a joint model that combines an autoregressive transformer decoder with a bidirectional transformer encoder in a single model with shared parameters. Upon training, we alternate between using the model as a decoder and as an encoder, with either a causal or bidirectional self-attention mechanism, alleviating problems typical for joint models. We evaluate the generative and predictive performance, as well as synergistic benefits of joint learning using HYFORMER across a variety of molecular tasks (Wu et al., 2018; Brown et al., 2019; Steshin, 2023; Chen et al., 2023). Our contributions are:

1. We propose a novel joint model, HYFORMER, that unifies the generative and the predictive task in a single set of parameters.

2. We demonstrate the synergistic benefits of joint modeling, where HYFORMER outperforms baselines on (i) conditional molecule generation, (ii) out-of-distribution property prediction and (iii) molecular representation learning via probing.

3. We show that HYFORMER rivals the generative and predictive performance of state-of-the-art purely generative and predictive models.

4. We showcase the applicability of joint modeling in a real-world drug design use case of discovering novel antimicrobial peptides.

# 2 Related Work

**Molecule Generation**  Existing generative approaches can be categorized into sequence- and graph-based models. Sequence-based methods represent molecules as SMILES (Weininger, 1988) or SELFIES (Krenn et al., 2020) and process tokenized strings using recurrent or transformer-based language models (Segler et al., 2018; Flam-Shepherd et al., 2022; Bagal et al., 2022). In contrast, graph-based models treat molecules as graphs and have been implemented using variational autoencoders (Liu et al., 2018; Jin et al., 2019; Maziarz et al., 2022; Hetzel et al., 2023), normalizing flows (Luo et al., 2021), energy-based models (Liu et al., 2021a), and graph transformers (Gao et al., 2024b). More recently, 3D-based generative models have been proposed to capture the spatial geometry of molecules (Hoogeboom et al., 2022; Guan et al., 2023; Gao et al., 2024a), however real world drug discovery pipelines continue to rely predominantly on 2D-molecular representations (Xiang et al., 2024).

**Molecular Property Prediction**   Analogously, prediction models leverage distinct molecular representations. Methods based on pre-trained language models predominantly work with SMILES (Wang et al., 2019; Fabian et al., 2020; Irwin et al., 2022; Sultan et al., 2024), while other approaches represent molecules as graphs (Li et al., 2021; Wang et al., 2022). Recent methods leverage the three-dimensional spatial structure of a molecule, either using graph neural networks (Fang et al., 2022) or transformers (Zhou et al., 2023). Finally, Yang et al. (2019); Fabian et al. (2020); Stokes et al. (2020) incorporate pre-computed physicochemical descriptors of molecules into training.

**Joint Models for Molecules**   Early joint models combine variational autoencoders with latent-space predictors (Gómez-Bombarelli et al., 2018; Maziarz et al., 2022). Regression Transformer (Born & Manica, 2023) frames property prediction as conditional sequence generation, but lacks unconditional generative capability. Graph2Seq (Gao et al., 2024b) is a graph-based encoder-decoder transformer, trained separately as a generative or as a predictive model, but evaluated on both molecule generation and property prediction. UniGEM (Feng et al., 2024) is a diffusion-based model for unified generation and prediction, however specializing in 3D molecular modeling and not directly applicable to standard SMILES-based benchmarks.

Therefore, the question of whether the transformer architecture can be used to implement a joint model for both SMILES-based generation and prediction, while enjoying synergistic benefits, remains open.

## 3   Background

**Problem Formulation**   The aim of *joint modeling* is to learn the joint distribution of the data and its properties $p(\mathbf{x}, y)$, i.e., to identify a model that at the same time generates new data and predicts its properties. We assume access to a *labeled dataset* $\mathcal{D} = \{(\mathbf{x}_n, y_n)\}_{n=1}^{N}$, sampled from the joint data distribution $p(\mathbf{x}, y)$, often accompanied with an *unlabeled dataset* $\mathcal{D}_U = \{\mathbf{x}_n\}_{n=1}^{N_U}$, sampled from $p(\mathbf{x})$. Here, examples $\mathbf{x}$ can be thought of as molecules and labels $y$ as molecular properties.

In the general formulation of Lasserre et al. (2006), joint modeling aims to learn the joint distribution $p(\mathbf{x}, y)$ by defining a *joint model* $p_{\theta,\phi}(\mathbf{x}, y)$ that factorizes into a *generative model* $p_\theta(\mathbf{x})$ and a *predictive model* $p_\phi(y \mid \mathbf{x})$ such that

$$p_{\theta,\phi}(\mathbf{x}, y) = p_\phi(y \mid \mathbf{x})p_\theta(\mathbf{x}), \tag{1}$$

where $\theta$ denotes the parameters of the generative model, and $\phi$ the parameters of the predictive model. Training of the joint model is equivalent to minimizing the negative log-likelihood, i.e., the *joint loss*

$$\ell_\lambda(\theta, \phi) = -\mathbb{E}_{(\mathbf{x},y)\sim p(\mathbf{x},y)}[\ln p_\theta(\mathbf{x}) + \lambda \ln p_\phi(y \mid \mathbf{x})], \tag{2}$$

where $\lambda \in \mathbb{R}$ weights the predictive and the generative parts.

Choosing the extent to which parameters $\theta$ and $\phi$ are shared and the way the joint loss is optimized, is crucial for obtaining a model with both a high generative and predictive performance, at the same time maintaining the synergistic benefits of joint learning (Lasserre et al., 2006).

### 3.1   Transformer-based Models

Transformers (Vaswani et al., 2017) achieve state-of-the-art performance in both molecule generation (Bagal et al., 2022) and property prediction (Zhou et al., 2023) tasks.

**Transformer Encoders and Decoders**   Transformers used for generation and for property prediction differ in the use of the *self-attention* mechanism. Transformer decoders, used for generative tasks, employ a *causal self-attention*

$$Att_\rightarrow(\mathbf{Q}, \mathbf{K}, \mathbf{V}) = \mathrm{softmax}\left(\frac{\mathbf{Q}\,\mathbf{K}^T}{\sqrt{d}} + \mathbf{M}_\rightarrow\right)\mathbf{V}, \tag{3}$$

where $\mathbf{Q}, \mathbf{K}, \mathbf{V} \in \mathbb{R}^{T \times d}$ are *query*, *key* and *value* matrices, respectively, $\mathbf{M}_\rightarrow \in \mathbb{R}^{T \times T}$ is a *causal mask*, i.e., a matrix such that $(\mathbf{M}_\rightarrow)_{ij} = 0$, if $i \geq j$, and $(\mathbf{M}_\rightarrow)_{ij} = -\infty$, otherwise, $T$ is the sequence length and $d$ is the

head dimension.[1] On the other hand, transformer encoders, used for predictive tasks, employ a *bidirectional self-attention*

$$Att_\leftrightarrow(\mathbf{Q}, \mathbf{K}, \mathbf{V}) = \text{softmax}\left(\frac{\mathbf{Q}\mathbf{K}^T}{\sqrt{d}} + \mathbf{M}_\leftrightarrow\right)\mathbf{V}, \tag{4}$$

where $\mathbf{M}_\leftrightarrow \in \mathbb{R}^{T \times T}$ is a *bidirectional mask*, i.e., $(\mathbf{M}_\leftrightarrow)_{ij} = 0$ for all $i, j \in [T]$.

**Alternating attention**    The definition of the transformer decoder and encoder can be generalized by using an alternating attention scheme (Dong et al., 2019):

$$Att_{\texttt{ATT\_Type}}(\mathbf{Q}, \mathbf{K}, \mathbf{V}) = \text{softmax}\left(\frac{\mathbf{Q}\mathbf{K}^T}{\sqrt{d}} + \mathbf{M}_{\texttt{ATT\_Type}}\right)\mathbf{V}, \tag{5}$$

where $\texttt{ATT\_Type} \in \{\rightarrow, \leftrightarrow\}$ and $\mathbf{M}_{\texttt{ATT\_Type}} = \mathbf{M}_\rightarrow$ is a causal mask upon using the model as a transformer decoder and $\mathbf{M}_{\texttt{ATT\_Type}} = \mathbf{M}_\leftrightarrow$, otherwise.

**Training transformers**    Training transformers proceeds in a two-step manner, by first *pre-training* the model on an unlabeled dataset and then *fine-tuning* the pre-trained model on a downstream task. Transformer decoders and encoders are pre-trained using different losses.

**Pre-training**    Transformer decoders, optimized for generative performance, are predominantly pre-trained using the negative log-likelihood loss $-\mathbb{E}_{\mathbf{x} \sim p(\mathbf{x})}[\ln p_\theta(\mathbf{x})]$. As the causal mask induces a factorization of the transformer decoder into an autoregressive model $p_\theta(\mathbf{x}) = \prod_{t=1}^T p_\theta(x_t \mid \mathbf{x}_{<t})$, where $\mathbf{x} = (x_1, \dots, x_T)$, the generative loss reduces to the *language modeling* (LM) loss

$$\ell_{\text{LM}}(\theta) = -\mathbb{E}_{\mathbf{x} \sim p(\mathbf{x})}\left[\sum_{t=1}^T \ln p_\theta(x_t \mid \mathbf{x}_{<t})\right]. \tag{6}$$

On the other hand, transformer encoders are usually pre-trained using *masked language modeling* (MLM) loss

$$\ell_{\text{MLM}}(\theta) = -\mathbb{E}_{\mathbf{x} \sim p(\mathbf{x})}\mathbb{E}_\mathcal{M}\left[\ln p_\theta(\mathbf{x}_\mathcal{M} \mid \mathbf{x}_\mathcal{R})\right], \tag{7}$$

where $\mathbf{x} = (x_1, \dots, x_T)$, $\mathcal{M}$ is a set of indices drawn uniformly at random from the set of token indices $\{1, \dots, T\}$ and the set of all tokens whose indices belongs to $\mathcal{M}$ are *masked tokens* $\mathbf{x}_\mathcal{M}$. The rest of the tokens $\mathbf{x}_\mathcal{R}$ are defined such that $\mathbf{x} = \mathbf{x}_\mathcal{M} \cup \mathbf{x}_\mathcal{R}$.

**Fine-tuning**    Next, the pretrained model is fine-tuned by defining a predictive head on top of the pretrained model and training it as a predictor on a labeled dataset using the *prediction loss*

$$\ell_{\text{PRED}}(\phi) = -\mathbb{E}_{(\mathbf{x}, y) \sim p(\mathbf{x}, y)}[\ln p_\phi(y \mid \mathbf{x})]. \tag{8}$$

## 4   Hyformer

We propose HYFORMER, a joint transformer-based model that unifies a generative decoder with a predictive encoder in a single set of shared parameters, using an alternating training scheme.

### 4.1   Model Formulation

HYFORMER unifies a decoder with an encoder using a transformer backbone $f_\theta(\mathbf{x}; \texttt{[TASK]})$ conditioned on a *task token* $\texttt{[TASK]} \in \{\texttt{[LM]}, \texttt{[PRED]}, \texttt{[MLM]}\}$. The task token facilitates switching between respective losses during

---

[1]We assume that the dimensions of the query, key, and value matrices are equal.

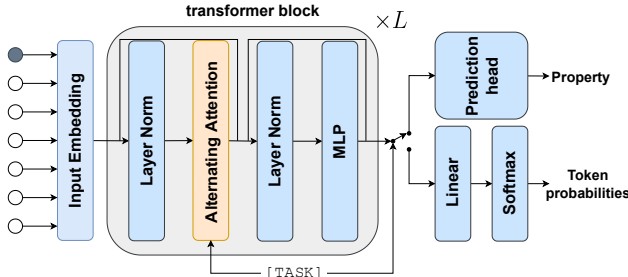

Figure 1: A schematic representation of HYFORMER. Depending on the task token [TASK], HYFORMER uses either a causal or a bidirectional mask, outputting token probabilities or predicted property values.

training (see Section 4.2) and determines whether the backbone $f_\theta$ processes input $\mathbf{x}$ in an autoregressive manner using a causal, or a bidirectional mask

$$\texttt{ATT\_Type} = \begin{cases} \rightarrow & \text{if } \texttt{[TASK]} = \texttt{[LM]}, \\ \leftrightarrow & \text{if } \texttt{[TASK]} \in \{\texttt{[PRED]}, \texttt{[MLM]}\}. \end{cases}$$

Finally, the generative $p_\theta(\mathbf{x})$ and predictive $p_\theta(y \mid \mathbf{x})$ parts of the joint model, factorized as

$$p_\theta(\mathbf{x}, y) := p_\theta(\mathbf{x})p_\theta(y \mid \mathbf{x}), \tag{9}$$

are implemented by adding a generative and a predictive head on the top of the shared backbone $f_\theta$.

---

**Algorithm 1** Training of HYFORMER
___
**Input:** Dataset $\mathcal{D}$ (labeled or unlabeled); model parameters $\theta$; task probabilities $\mathbf{p}_{\texttt{[TASK]}}$.
    For pre-training: $\texttt{[TASK]} \in \{\texttt{[LM]}, \texttt{[PRED]}, \texttt{[MLM]}\}$, for fine-tuning: $\texttt{[TASK]} \in \{\texttt{[LM]}, \texttt{[PRED]}\}$.
 1: **while** stopping criterion not met **do**
 2:    Sample task $\texttt{[TASK]} \sim \text{CAT}(\mathbf{p}_{\texttt{[TASK]}})$
 3:    Select loss $\ell_{\texttt{[TASK]}}$ and the corresponding attention mask
 4:    Update model parameters $\theta$ using the gradient of $\ell_{\texttt{[TASK]}}$
 5: **end while**

---

## 4.2 Hyformer Training

As with standard transformer-based models, the training of HYFORMER is divided into a pre-training and a fine-tuning stage.

**Joint Pre-training**   To unify the generative and the predictive functionalities in a single model, we pre-train HYFORMER using a variant of the joint loss (Eq. 2). For the generative part, we use the language modeling loss $\ell_{\text{LM}}$, while for the predictive part, we use the masked language modeling loss $\ell_{\text{MLM}}$ and the predictive loss $\ell_{\text{PRED}}$, with the combined loss being defined as:

$$\ell_{\text{HYFORMER}} = \ell_{\text{LM}} + \mu\ell_{\text{MLM}} + \eta\ell_{\text{PRED}}. \tag{10}$$

As *pre-training labels*, we use values analytically computable from the input sequences, e.g., molecular descriptors, such as molecular weight for small molecules, or hydrophobicity for peptides. When the pre-training labels are not available, HYFORMER is pre-trained without the predictive loss $\ell_{\text{PRED}}$. Analogously to multitask learning (Raffel et al., 2023), the weighted loss $\ell_{\text{HYFORMER}}$ (Eq. 10) is effectively implemented using a vector of task probabilities $\mathbf{p}_{\texttt{[TASK]}} = (p_{\texttt{[LM]}}, p_{\texttt{[MLM]}}, p_{\texttt{[PRED]}})$, which defines how the generative and predictive capabilities of the joint model are balanced.

During training, the shared parameters $\theta$ are updated differently depending on the task token. If $\texttt{[TASK]} \in \{\texttt{[PRED]}, \texttt{[MLM]}\}$, a bidirectional attention mask $\mathbf{M}_{\leftrightarrow}$ is applied and all attention module weights are updated, since the bidirectional mask does not restrict information flow. Conversely, if $\texttt{[TASK]} = \texttt{[LM]}$, a causal mask $\mathbf{M}_{\rightarrow}$ is applied, restricting each token to attend only to its left context, altering the gradients of the attention module, due to the functional form of the Jacobian of the softmax function, alleviating gradient interference typical for joint modeling (Appendix D.1).

**Fine-tuning**   We fine-tune HYFORMER using the joint loss (Eq. 2), defined as

$$\ell_{\text{HYFORMER}} = \ell_{\text{LM}} + \lambda \ell_{\text{PRED}}. \tag{11}$$

Analogously to pre-training, HYFORMER alternates between the generative and predictive task, to balance their objectives, based on a pre-defined vector of task probabilities $\mathbf{p}_{\texttt{[TASK]}} = (p_{\texttt{[LM]}}, p_{\texttt{[PRED]}})$. We assume that *fine-tuning labels* used in loss $\ell_{\text{PRED}}$ are different than in the pre-training phase and are defined by the downstream prediction task. Specifically, we omit the masked language modeling loss, to focus on the downstream task while retaining the generative capabilities of the model.

## 4.3   Sampling

Sampling from HYFORMER exploits the generative $p_\theta(\mathbf{x})$ and predictive part $p_\theta(y \mid \mathbf{x})$ depending on the sampling mode: unconditional or conditional.

**Unconditional Generation**   In unconditional generation, we sample $\mathbf{x} \sim p_\theta(\mathbf{x})$ using the autoregressive part of the model. This addresses a limitation of conditionally trained generative models (Bagal et al., 2022) and joint models trained without a pure unsupervised objective (Born & Manica, 2023), where generating a single example requires conditioning on a fixed property value inferred from a dataset.

**Conditional Generation**   To generate $(\mathbf{x}, y) \sim p_\theta(\mathbf{x}, y)$ that satisfies a condition $Y \subseteq \mathcal{Y}$, HYFORMER samples $K$-many examples $\mathbf{x}_1, \ldots, \mathbf{x}_K \sim p_\theta(\mathbf{x})$ and, for every $k = 1, \ldots, K$, accepts sample $\mathbf{x}_k$, if the predictor $p_\theta(y \mid \mathbf{x})$ classifies $\mathbf{x}_k$ as having property $Y$. As a simple consequence of the Bayes rule, the above procedure yields a correct conditional sampling procedure, as

$$p(\mathbf{x} \mid y \in Y) \propto p(y \in Y \mid \mathbf{x}) p(\mathbf{x}), \tag{12}$$

for $y \in Y \subseteq \mathcal{Y}$ such that $p(y \in Y) > 0$. Note that the conditional sampling procedure of HYFORMER is a variant of best-of-$K$ sampling, a provably near-optimal solution to the KL-regularized RL problem (Yang et al., 2019) that has been shown to outperform other conditional sampling methods for LLMs, including state-of-the-art reinforcement learning methods like PPO and DPO (Touvron et al., 2023; Mudgal et al., 2023; Gao et al., 2023; Rafailov et al., 2023). Crucially, HYFORMER leverages a jointly trained predictor $p_\theta(y \mid x)$ over a unified representation space, resulting in tighter alignment between generation and control. This coherence is particularly valuable in drug discovery, where the primary objective is not throughput, but *precision and sample efficiency*, that is, generating a small number of high-quality candidates with minimal false positives.

## 5   Experiments

We evaluate HYFORMER across a broad range of molecular modeling tasks. First, we demonstrate the synergistic benefits of joint modeling in three settings: (i) conditional generation on GuacaMol dataset (Brown et al., 2019), (ii) out-of-distribution (OOD) property prediction on Hit Identification task from the Lo-Hi benchmark (Steshin, 2023) and (iii) representation learning via probing on MoleculeNet benchmark (Wu et al., 2018). Subsequently, we show that HYFORMER rivals state-of-the-art generative and predictive models in both unconditional generation on GuacaMol and property prediction on MoleculeNet. Finally, we apply HYFORMER to antimicrobial peptide (AMP) design, showcasing the benefits of our joint modeling approach. Experimental details and additional results are provided in Appendix G, H and I.

Table 1: Conditional generative performance on GuacaMol dataset. Best model is marked **bold**.

| MODEL | JOINT | METRIC | QED | SA | LOGP | AVG. |
|---|---|---|---|---|---|---|
| MOLGPT | ✗ | MAD ↓ | 0.087 | 0.019 | 0.276 | 0.127 |
| | | SD ↓ | 0.074 | 0.017 | 0.262 | 0.118 |
| | | VALIDITY ↑ | 0.985 | 0.986 | 0.982 | 0.984 |
| GRAPHGPT | ✗ | MAD ↓ | 0.039 | 0.011 | 0.158 | 0.069 |
| | | SD ↓ | 0.082 | 0.047 | 0.653 | 0.261 |
| | | VALIDITY ↑ | **0.998** | **0.997** | **0.992** | **0.995** |
| HYFORMER | ✗ | MAD ↓ | 0.031 (0.003) | 0.015 (0.001) | 0.131 (0.010) | 0.059 (0.004) |
| | | SD ↓ | 0.045 (0.004) | 0.020 (0.001) | 0.170 (0.014) | 0.078 (0.006) |
| | | VALIDITY ↑ | 0.993 (0.003) | 0.990 (0.004) | 0.985 (0.014) | 0.989 (0.007) |
| | ✓ | MAD ↓ | **0.008 (0.001)** | **0.005 (0.000)** | **0.041 (0.002)** | **0.018 (0.001)** |
| | | SD ↓ | **0.015 (0.002)** | **0.009 (0.002)** | **0.051 (0.004)** | **0.025 (0.003)** |
| | | VALIDITY ↑ | 0.990 (0.007) | 0.985 (0.003) | 0.987 (0.006) | 0.987 (0.005) |

## 5.1 Synergistic Benefits of Hyformer

### 5.1.1 Conditional Molecule Generation

To demonstrate the synergistic benefits of HYFORMER in generating molecules with specific molecular properties, we follow the setup of Bagal et al. (2022) and jointly pre-train HYFORMER scaled to 8.5M parameters on GuacaMol dataset with 1.3M molecules, using pre-computed molecular descriptors (Yang et al., 2019). We subsequently jointly fine-tune HYFORMER on GuacaMol dataset with QED, SA, and LogP molecular properties, as fine-tuning labels, and generate molecules with specific properties using HYFORMER's conditional sampling procedure. Pre-training and experimental details alongside results for all property settings can be found in Appendix G and H.1.

Following (Gao et al., 2024b), we compare HYFORMER to MolGPT (Bagal et al., 2022) and GraphGPT (Gao et al., 2024b) using: mean absolute deviation (MAD) from the target property value, standard deviation (SD) of the generated property values and validity of the generated molecules. Evaluation is averaged across three target values per each property: QED:{0.5, 0.7, 0.9}, SA:{0.7, 0.8, 0.9}, and logP:{0.0, 2.0, 4.0}. Additionally, we compare to a non-joint variant of HYFORMER, in which the predictive head is fine-tuned with prediction loss, on top of a frozen, pre-trained generative part, i.e., without joint fine-tuning.

The jointly fine-tuned HYFORMER achieves the lowest MAD and SD across all properties, while maintaining high validity, outperforming all baselines. Notably, HYFORMER improves controllability over it's non-joint counterpart, confirming that joint fine-tuning enhances conditional generation. Although GraphGPT attains slightly higher validity, it does so at the cost of reduced controllability. These results demonstrate that joint modeling enables robust property-conditioned molecular generation across a range of chemically relevant targets.

### 5.1.2 Out-of-Distribution Molecular Property Prediction

To evaluate the ability of HYFORMER to predict molecular properties in an out-of-distribution (OOD) setting, we jointly pre-train HYFORMER scaled to 50M parameters on 19M molecules from (Zhou et al., 2023), together with pre-computed molecular descriptors (Yang et al., 2019), and benchmark on the Hit Identification (Hi) task from the Lo-Hi benchmark (Steshin, 2023). The Hi task requires generalization to molecular scaffolds not seen during training, with the test set constructed such that no molecule has a Tanimoto similarity greater than 0.4 (based on ECFP4 fingerprints) to any molecule in the training set. This setup mimics realistic drug discovery scenarios, where generalization beyond known chemical space is essential. For experimental details, see Appendix G and H.2.

We follow the setup of (Steshin, 2023) and compare jointly fine-tuned HYFORMER to all models reported in (Steshin, 2023); machine learning models: k-NN, gradient boosting (GB), SVM and MLP, trained on molecular fingerprints (ECFP4, MACCS) and deep learning models: Chemformer (Yang et al., 2019) and Graphformer (Ying et al., 2021; Shi et al., 2022). Moreover, we compare to HYFORMER (no-joint), which is a version of our model pre-trained using MLM loss, hence without alternating attention, and fine-tuned using the prediction loss only.

Table 2: Predictive performance (AUPRC) on Hit Identification (Hi) task from Lo-Hi benchmark. Mean and standard deviation across 3 random seeds.

| | Dataset, AUPRC (↑) | | | |
|---|---|---|---|---|
| Model | DRD2-Hi | HIV-Hi | KDR-Hi | Sol-Hi |
| Dummy baseline | 0.677 (0.061) | 0.040 (0.014) | 0.609 (0.081) | 0.215 (0.008) |
| KNN (ECFP4) | 0.706 (0.047) | 0.067 (0.029) | 0.646 (0.048) | 0.426 (0.022) |
| KNN (MACCS) | 0.702 (0.042) | 0.072 (0.036) | 0.610 (0.072) | 0.422 (0.009) |
| GB (ECFP4) | 0.736 (0.050) | 0.080 (0.038) | 0.607 (0.067) | 0.429 (0.006) |
| GB (MACCS) | 0.751 (0.063) | 0.058 (0.030) | 0.603 (0.074) | 0.502 (0.045) |
| SVM (ECFP4) | 0.677 (0.061) | 0.040 (0.014) | 0.611 (0.081) | 0.298 (0.047) |
| SVM (MACCS) | 0.713 (0.050) | 0.042 (0.015) | 0.605 (0.082) | 0.308 (0.021) |
| MLP (ECFP4) | 0.717 (0.063) | 0.049 (0.019) | 0.626 (0.047) | 0.403 (0.017) |
| MLP (MACCS) | 0.696 (0.048) | 0.052 (0.018) | 0.613 (0.077) | 0.462 (0.048) |
| Chemprop | 0.782 (0.062) | 0.148 (0.114) | 0.676 (0.026) | 0.618 (0.030) |
| Graphormer | 0.729 (0.039) | 0.096 (0.070) | - | - |
| Hyformer (no-joint) | 0.778 (0.070) | 0.154 (0.108) | 0.675 (0.046) | 0.601 (0.040) |
| Hyformer | 0.784 (0.082) | 0.158 (0.128) | 0.701 (0.022) | 0.640 (0.036) |

HYFORMER achieves the highest mean AUPRC across all datasets (Table 2), outperforming fingerprint-based baselines and demonstrating the potential of deep learning methods in real-world drug discovery. The consistent ranking in favor of HYFORMER shows the benefits of joint modeling in out-of-distribution molecular property prediction, although the differences are not statistically significant at the 95% confidence level.

### 5.1.3 Molecular Representation Learning

Table 3: Molecular representation learning performance of predictive, generative and joint models on MoleculeNet benchmark, evaluated using linear and KNN probing. Best model within each probing method is marked **bold**.

| | Type | Model | Dataset, RMSE ↓ | | | Dataset, AUCROC ↑ | | | | | | |
|---|---|---|---|---|---|---|---|---|---|---|---|---|
| | | | Esol | Freesolv | Lipo | BBBP | BACE | ClinTox | Tox21 | ToxCast | SIDER | HIV |
| LINEAR | P. | Uni-Mol | 1.350 | **2.503** | 1.002 | 65.5 | 66.3 | 74.3 | 70.1 | 59.9 | 58.1 | 73.6 |
| | P. | Hyformer (no-joint) | 1.256 | 2.640 | 0.894 | 68.4 | 73.6 | 98.8 | **73.4** | **61.2** | 58.8 | 75.9 |
| | G. | MolGPT | 1.299 | 4.110 | 1.033 | 66.8 | 79.1 | 97.8 | 71.9 | 60.5 | 59.2 | **77.5** |
| | J. | MoLeR | **1.223** | 4.935 | 0.938 | 67.8 | **79.5** | 84.6 | 71.1 | 59.3 | 58.3 | 74.6 |
| | J. | RT | 2.510 | 4.515 | 1.158 | 54.7 | 63.1 | 57.3 | 50.5 | 52.8 | 54.5 | 65.6 |
| | J. | Graph2Seq | 1.498 | 3.486 | 0.890 | 66.0 | 76.7 | 72.0 | 71.2 | 60.4 | 50.5 | 57.1 |
| | J. | Hyformer | 1.527 | 4.294 | **0.887** | **68.5** | 77.2 | **99.5** | 72.4 | 60.7 | **60.8** | 74.7 |
| KNN | P. | Uni-Mol | 1.579 | 3.403 | 1.025 | 60.0 | 75.9 | 78.0 | 64.7 | 57.5 | 61.0 | 64.3 |
| | P. | Hyformer (no-joint) | 1.380 | 3.254 | 0.978 | 67.8 | 75.4 | 89.0 | 66.3 | 57.6 | 58.1 | 71.4 |
| | G. | MolGPT | **1.232** | **3.075** | 0.987 | 68.4 | 71.9 | **94.2** | 66.0 | 56.9 | 61.0 | 70.5 |
| | J. | MoLeR | 1.802 | 4.061 | 1.096 | 59.4 | 72.0 | 71.2 | 64.9 | 53.3 | 57.3 | 67.3 |
| | J. | RT | 2.411 | 4.734 | 1.242 | 59.3 | 56.1 | 59.4 | 50.8 | 52.2 | 51.2 | 54.1 |
| | J. | Graph2Seq | 1.361 | 3.796 | 0.967 | **71.0** | **80.6** | 56.3 | 67.7 | 57.8 | 49.9 | 52.4 |
| | J. | Hyformer | 1.260 | 3.999 | **0.902** | 69.5 | 78.4 | 93.8 | **71.2** | **59.3** | **64.1** | **71.8** |

To assess the quality of molecular representations learned by HYFORMER, we introduce a novel probing protocol that emulates a typical drug discovery setting, where fixed molecular embeddings are used as inputs to downstream predictive models. In this setup, we train simple linear models with L2 regularization, and k-nearest neighbor (KNN) predictors on the top of frozen embeddings extracted from the respective pre-trained models. To ensure comparability with MoleculeNet benchmark (Section 5.2.2), we reuse the same datasets, data splits, and model checkpoints. Implementation details are provided in Appendix H.3.

We compare representations extracted from jointly pre-trained HYFORMER to those extracted from a range of baselines, including state-of-the-art generative (MolGPT (Bagal et al., 2022)), predictive (Uni-Mol (Zhou et al., 2023)), and joint models: MoLeR (Maziarz et al., 2022), Regression Transformer (RT) (Born & Manica, 2023) and Graph2Seq (Gao et al., 2024b). Moreover, to quantify the effect of alternating attention and joint pre-training, we compare to HYFORMER (no-joint), the version of our model trained solely with MLM loss.

The jointly pre-trained representations from HYFORMER are the most predictive across both KNN and linear probings, achieving the best performance on 4 out of 10 datasets for linear, and 5 out of 10 datasets for KNN, outperforming all other baselines (Table 3). An additional analysis of linear probing on ClinTox shows

Table 4: Unconditional generative performance on GuacaMol distribution learning benchmarks. The best model in each category is marked **bold**.

| MODEL | FCD SCORE ↑ | KL DIV. SCORE ↑ | VAL. ↑ | UNIQ. ↑ | NOV. ↑ |
|---|---|---|---|---|---|
| *GRAPH-BASED* | | | | | |
| JT-VAE | 0.750 | 0.940 | **1.000** | - | - |
| MOLER | 0.625 | 0.964 | **1.000** | **1.000** | **0.991** |
| MAGNET | **0.760** | 0.950 | **1.000** | - | - |
| MICAM | 0.731 | **0.989** | **1.000** | 0.994 | 0.986 |
| *SMILES-BASED* | | | | | |
| VAE | 0.863 | 0.982 | 0.870 | 0.999 | 0.974 |
| LSTM | 0.913 | 0.991 | 0.959 | **1.000** | 0.912 |
| MOLGPT | 0.907 | 0.992 | 0.981 | 0.998 | **1.000** |
| HYFORMER$_{\tau=0.9}$ | 0.897 (0.002) | **0.995 (0.000)** | **0.986 (0.001)** | 0.999 (0.000) | 0.879 (0.006) |
| HYFORMER$_{\tau=1.0}$ | **0.918 (0.002)** | 0.989 (0.001) | 0.978 (0.000) | 0.999 (0.000) | 0.908 (0.002) |
| HYFORMER$_{\tau=1.1}$ | 0.894 (0.002) | 0.977 (0.001) | 0.965 (0.001) | **1.000 (0.000)** | 0.931 (0.001) |

high per-target F1 scores of 0.98 and 0.90, indicating robust performance of HYFORMER across targets. The next best models, Hyformer (no-joint) and MoLeR for linear and MolGPT for KNN probing, rank first on 2 and 3 out of 10 datasets, respectively. Notably, joint models outperform UniMol, the state-of-the-art property predictor, on all datasets, except for Freesolv with linear probing, highlighting the effectiveness of joint modeling for transferable molecular representation learning.

## 5.2 Generative and predictive performance of Hyformer

We next confirm that HYFORMER effectively addresses the challenges of joint training, while it enjoys the synergistic benefits described above, it does not sacrifice generative or predictive performance compared to state-of-the-art models trained separately for these tasks.

### 5.2.1 Unconditional Molecule Generation

To evaluate the unconditional generative performance of HYFORMER, we perform an evaluation on the Guacamol distribution learning benchmark (Brown et al., 2019). We use HYFORMER scaled to 8.5M parameters and trained on GuacaMol dataset with 1.3M molecules, together with pre-computed molecular descriptors (Yang et al., 2019), and investigate the impact of sampling temperature $\tau$. For experimental details, see Appendix H.4.

We compare to state-of-the-art unconditional generative models; SMILES-based: VAE (Kingma & Welling, 2013), LSTM (Gers & Schmidhuber, 2001), MolGPT (Bagal et al., 2022) and graph-based: JT-VAE (Jin et al., 2019), MoLeR (Maziarz et al., 2022), MAGNet (Hetzel et al., 2023), MiCaM (Geng et al., 2023). We omit RT (Born & Manica, 2023) and GraphGPT (Gao et al., 2024b) as they do not generate molecules unconditionally or provide results on the GuacaMol benchmark.

HYFORMER, with top FCD and KL div. score values, outperforms graph-based models, while achieving the highest validity among SMILES-based models. Across various sampling temperatures $\tau$, HYFORMER consistently lies on the Pareto front, balancing distributional fidelity (FCD Score, KL div. Score), validity and uniqueness. Overall, SMILES-based models outperform those based on theoretically more informative graph representations in terms of FCD Score, at the expense of not always sampling valid molecules.

### 5.2.2 Molecular Property Prediction

To evaluate the predictive performance of HYFORMER, we use HYFORMER scaled to 50M parameters on 19M molecules from (Zhou et al., 2023), together with pre-computed molecular descriptors (Yang et al., 2019), and fine-tune end-to-end on MoleculeNet benchmark (Wu et al., 2018). For experimental details, see Appendix H.5.

We follow the experimental protocol of (Zhou et al., 2023), use scaffold splitting and compare to predictive models: D-MPNN (Yang et al., 2019), AttentiveFP (Xiong et al., 2019), N-gram (Liu et al., 2019) with Random Forest and XGBoost (Chen & Guestrin, 2016), PretrainGNN (Hu et al., 2019), GROVER (Rong et al., 2020), MolCLR (Wang et al., 2022), Mole-BERT (Xia et al., 2023), GraphMVP (Liu et al., 2021b),

Table 5: Predictive performance of predictive and joint models on the MoleculeNet benchmark. Mean and standard deviation across 3 random seeds. The best model in each category, statistically significant at the 95% confidence level, is marked **bold**.

| | MODEL | DATASET, RMSE ↓ | | | DATASET, AUCROC ↑ | | | | | | |
|---|---|---|---|---|---|---|---|---|---|---|---|
| | | ESOL | FREESOLV | LIPO | BBBP | BACE | CLINTOX | TOX21 | TOXCAST | SIDER | HIV |
| PREDICTIVE | D-MPNN | 1.050(0.008) | 2.082(0.082) | 0.683(0.016) | 71.0(0.3) | 80.9(0.6) | 90.6(0.6) | 75.9(0.7) | 65.5(0.3) | 57.0(0.7) | 77.1(0.5) |
| | ATTENTIVE FP | 0.877(0.029) | 2.073(0.183) | 0.721(0.001) | 64.3(1.8) | 78.4(0.02) | 84.7(0.3) | 76.1(0.5) | 63.7(0.2) | 60.6(3.2) | 75.7(1.4) |
| | N-GRAMRF | 1.074(0.107) | 2.688(0.085) | 0.812(0.028) | 69.7(0.6) | 77.9(1.5) | 77.5(4.0) | 74.3(0.4) | - | 66.8(0.7) | 77.2(0.1) |
| | N-GRAMXGB | 1.083(0.082) | 5.061(0.744) | 2.072(0.030) | 69.1(0.8) | 79.1(1.3) | 87.5(2.7) | 75.8(0.9) | - | 65.5(0.7) | 78.7(0.4) |
| | PRETRAINGNN | 1.100(0.006) | 2.764(0.002) | 0.739(0.003) | 68.7(1.3) | 84.5(0.7) | 72.6(1.5) | 78.1(0.6) | 65.7(0.6) | 62.7(0.8) | 79.9(0.7) |
| | GROVERBASE | 0.983(0.090) | 2.176(0.052) | 0.817(0.008) | 70.0(0.1) | 82.6(0.7) | 81.2(3.0) | 74.3(0.1) | 65.4(0.4) | 64.8(0.6) | 62.5(0.9) |
| | GROVERLARGE | 0.895(0.017) | 2.272(0.051) | 0.823(0.010) | 69.5(0.1) | 81.0(1.4) | 76.2(3.7) | 73.5(0.1) | 65.3(0.5) | 65.4(0.1) | 68.2(1.1) |
| | GRAPHMVP | 1.029(0.033) | - | 0.681(0.010) | 72.4(1.6) | 81.2(0.9) | 79.1(2.8) | 75.9(0.5) | 63.1(0.4) | 63.9(1.2) | 77.0(1.2) |
| | MOLCLR | 1.271(0.040) | 2.594(0.249) | 0.691(0.004) | 72.2(2.1) | 82.4(0.9) | 91.2(3.5) | 75.0(0.2) | - | 58.9(1.4) | 78.1(0.5) |
| | MOLE-BERT | 1.015 (0.030) | - | 0.676 (0.017) | 71.9 (1.6) | 80.8 (1.4) | 78.9 (3.0) | 76.8 (0.5) | 64.3 (0.2) | - | - |
| | GEM | 0.798(0.029) | 1.877(0.094) | 0.660(0.008) | 72.4(0.4) | 85.6(1.1) | 90.1(1.3) | 78.1(0.1) | 69.2(0.4) | 67.2(0.4) | 80.6(0.9) |
| | UNI-MOL | 0.788(0.029) | **1.480(0.048)** | **0.603(0.010)** | 72.9(0.6) | 85.7(0.2) | 91.9(1.8) | **79.6(0.5)** | 69.6(0.1) | 65.9(1.3) | 80.8(0.3) |
| JOINT | GRAPH2SEQ | 0.860(0.024) | 1.797(0.237) | 0.716(0.019) | 72.8(1.5) | 83.4(1.0) | - | 76.9(0.3) | 65.4(0.5) | 68.2(0.9) | 79.4(3.9) |
| | HYFORMER | **0.774(0.026)** | 2.047(0.076) | **0.643(0.002)** | 75.9(0.9) | 83.8(1.1) | **99.2(0.5)** | **79.2(0.1)** | 65.5(0.6) | 65.7(1.6) | 80.0(1.0) |

GEM (Fang et al., 2022), UniMol (Zhou et al., 2023) and a joint model: Graph2Seq (Gao et al., 2024b). We omit RT (Born & Manica, 2023) and other models that use random splitting.

HYFORMER obtains the lowest RMSE on Esol, and highest AUROC on BBBP and ClinTox, outperforming all models on 3 out of 10 datasets (Table 5). Moreover, HYFORMER performs better than Graph2Seq, the only other joint model capable of simultaneous molecule generation and property prediction, on 8 out of 10 datasets. Altogether, HYFORMER outperforms the other joint learning model, Graph2Seq, and successfully rivals the performance of purely predictive models, demonstrating the efficiency of our joint learning strategy.

Table 6: Conditional generative performance on antimicrobial peptide design. Mean and standard deviation computed over 100 bootstrap iterations. The best model is marked **bold**.

| MODEL | PERPLEXITY[2] | DIVERSITY ↑ | FITNESS ↑ | HYDRAMP$_{MIC}$ ↑ | AMPLIFY ↑ | AMPEPPY ↑ |
|---|---|---|---|---|---|---|
| PEPCVAE | 20.11 (0.14) | **0.87** (0.0003) | 0.07 (0.0004) | 0.20 (0.0016) | 0.49 (0.0016) | 0.52 (0.0007) |
| AMPGAN | 18.58 (0.10) | 0.81 (0.0005) | 0.12 (0.0005) | 0.32 (0.0019) | 0.64 (0.0018) | 0.54 (0.0008) |
| HYDRAMP | 20.14 (0.12) | **0.86** (0.0004) | 0.09 (0.0005) | 0.49 (0.0021) | 0.59 (0.0016) | 0.52 (0.0006) |
| AMP-DIFFUSION | 16.93 (0.18) | 0.82 (0.0004) | 0.13 (0.0005) | 0.26 (0.0018) | 0.20 (0.0014) | 0.38 (0.0006) |
| HYFORMER | 17.98 (0.06) | 0.80 (0.0005) | **0.19** (0.0006) | **0.80** (0.0019) | **0.94** (0.0027) | **0.72** (0.0018) |

## 5.3 Antimicrobial Peptide Design

To show the benefits of joint learning in a real-world use case related to drug discovery, we apply HYFORMER to the task of antimicrobial peptide (AMP) design (Chen et al., 2023), i.e., generating AMPs with low minimal inhibitory concentration values (MIC) against *E. coli* bacteria. We jointly pre-train HYFORMER on 3.5M general-purpose peptide sequences, and subsequently on 1M AMP sequences, together with 39 physicochemical descriptors from *peptidy* package (Özçelik et al., 2025). Next, we jointly fine-tune HYFORMER on 4,547 peptides with their MIC values (Szymczak et al., 2023) and conditionally sample 50K peptides with an MIC regressor threshold set to $\leq 10^{0.3} \approx 2\,\mu\text{M}$. For experimental details, see Appendix H.6.

We compare HYFORMER AMP generation baselines: PepCVAE (Das et al., 2018), AMPGAN (Van Oort et al., 2021), HydrAMP (Szymczak et al., 2023), and AMP-Diffusion (Chen et al., 2023). Evaluation is based on four criteria: Perplexity (Torres et al., 2025), Diversity and Fitness (Li et al., 2024), and success rates in generating AMPs and low-MIC candidates. For the latter, we use HydrAMP$_{MIC}$, Amplify (Li et al., 2022), and amPEPpy (Lawrence et al., 2020) classifiers as state-of-the-art *in-silico* oracles.

HYFORMER outperforms all baseline models by a large margin in terms of generating peptides with a high fitness and AMP probability, as evaluated by all oracle classifiers (Table 6). Despite the stringent conditioning MIC threshold of $2\,\mu\text{M}$, HYFORMER maintains competitive perplexity and high diversity. These results suggest that even when constrained to explore less charted regions of sequence space, HYFORMER is able to generate biologically plausible and novel peptide candidates.

---

[2]We report perplexity, but do not seek to minimize it, as it inherently balances plausibility and novelty.

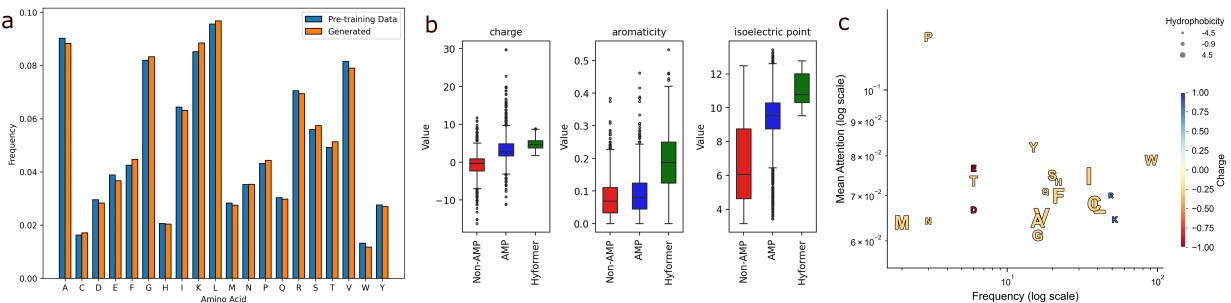

Figure 2: **(a)** Amino-acid distributions between the pre-training data and unconditionally generated sequences. **(b)** Distributions of charge, aromaticity, and isoelectric point (pI) for: non-AMP, AMP and conditionally generated sequences. **(c)** Frequency of crossing an attention threshold (x-axis) vs. mean attention weight (y-axis) for distinct amino-acids, colored by charge and sized by hydrophobicity.

To further validate the biological relevance of the generated peptides, we show that both unconditional sampling from jointly pre-trained HYFORMER, and conditional sampling from the fine-tuned model produces amino-acid distributions in close agreement with the training data (Figure 2a). Despite this very close agreement, the conditionally sampled peptides obtain a significant improvement of charge, aromaticity, and isoelectric point over the known non-AMPs, as compared to known AMPs (Fig. 2b). Finally, to gain insight into which amino acids contribute most to antimicrobial activity, we analyze the attention weights of HYFORMER (Fig. 2c). The attention mechanism frequently prioritizes highly charged Arginine (R) and Lysine (K), which is expected as high AMP activity is associated with increased charge. The high attention frequency on Tryptophan (W) agrees with previous reports about this amino-acid's unique ability to interact with the interface of the bacterial membrane (Bi et al., 2014). Finally, the high attention that HYFORMER puts on Proline (P) agrees with the known high potency of Proline-rich AMPs, which kill bacteria via a specific, non-lytic mechanism (Lai et al., 2019).

# 6 Discussion

In this paper, we introduced HYFORMER, a transformer-based joint model that combines an autoregressive decoder and a bidirectional encoder within a single set of shared parameters, using an alternating attention mechanism and joint pre-training. We showed that HYFORMER provides synergistic benefits in conditional sampling, representation learning and out-of-distribution property prediction, with ablations highlighting the specific contributions of alternating attention and joint training. Furthermore, we validated the utility of joint modeling in a real-world antimicrobial peptide design task. Our results indicate that HYFORMER successfully unifies molecular generation and property prediction for SMILES-based molecular representations, opening the avenue for the integration into real-world drug discovery pipelines, where informative molecular representations, robustness to OOD examples and robust conditional sampling are crucial.

**Limitations & Future Work** However, joint modeling introduces an inherent trade-off. While shared parameters promote synergistic benefits and learning unified representations, they may limit task-specific specialization. Therefore, a promising direction for future work is designing dynamic or modular attention architectures that allocate capacity across tasks more flexibly, while preserving synergistic benefits. Moreover, to ensure fair comparison with prior work and isolate the effect of joint learning, we deliberately restricted model scale and relied on a fixed set of analytically computed descriptors. The extent to which the observed synergistic benefits carry over to other modalities, such as 3D structures, morphology or transcriptomics, remains an open question.

## Acknowledgments

This project has received funding from the European Research Council (ERC) under the European Funding Union's Horizon 2020 research and innovation programme (grant agreement No 810115 – DOG-AMP).

We thank Hassan Akell for insightful discussions and careful review of the theoretical part of this paper. Their feedback substantially improved the clarity, rigor, and presentation of the theoretical analysis.

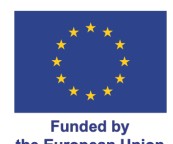 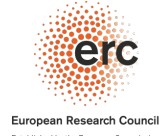

**Conflict of interest**   Projects at Ewa Szczurek Lab at the University of Warsaw are co-funded by Merck Healthcare GmbH.

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

## A  Impact Statement

The goal of this this work is to improve the field of deep generative modeling and, potentially, drug design. An example of potential malicious use of our approach would be training a deep generative model for generating new toxic molecules. However, the intention of this paper is to provide tools that will facilitate designing new potential medications.

## B  Extended Discussion

**Extended Novelty Statement**  The alternating self-attention scheme is closely related to prior multitask transformer work (e.g., Dong et al. (2019)). However, to the best of our knowledge, HYFORMER is the first model to employ an alternating attention scheme during both pre-training and fine-tuning, resulting in a joint model that unifies molecular generation and property prediction. In contrast, Dong et al. (2019) apply alternating attention only during pre-training. Moreover, HYFORMER explicitly combines reconstruction-based losses: LM and MLM, with a prediction loss (Eq. 10), while Dong et al. (2019) rely exclusively on reconstruction-based losses, without incorporating any supervision based on labeled data. Together, these architectural and objective-level differences enable joint generative and predictive modeling, and distinguish HYFORMER from prior alternating-attention-based models.

**Extended Future Work**  An interesting direction for future work is to cast molecular property prediction as a purely generative task. While such a formulation could further unify generation and prediction within a single modeling paradigm, it introduces nontrivial challenges, most notably the principled tokenization and representation of continuous molecular properties. Moreover, the proposed framework naturally extends to modeling molecular interactions. Since HYFORMER natively supports multimodal inputs, small-molecule–protein interactions can be incorporated by conditioning the decoder on protein embeddings and augmenting training with additional pretraining objectives. Exploring such extensions to jointly model multiple molecular modalities represents a promising avenue for future research.

## C  Notation

| Symbol | Meaning |
|---|---|
| $[N]$ | Set of integers $1, \ldots, N$ |
| $\mathbf{A}$ | Matrix |
| $\mathbf{A}^T$ | Transposed matrix $\mathbf{A}$ |
| $\mathbf{A}_i, \mathbf{A}_{ij}, \mathbf{A}^{ij}$ | Matrix indexed for some purpose |
| $(\mathbf{A})_i, \mathbf{A}[i], A_i$ | The $i$-th row of matrix $\mathbf{A}$ |
| $(\mathbf{A})_{ij}, \mathbf{A}[i,j], A_{ij}$ | The $i$-th, $j$-th entry of matrix $\mathbf{A}$ |
| $\mathbf{a}$ | Vector (column-vector) |
| $\mathbf{a}_i, \mathbf{a}_{ij}, \mathbf{a}^{ij}$ | Vector indexed for some purpose |
| $(\mathbf{a})_i, \mathbf{a}[i], a_i$ | The $i$-th entry of vector $\mathbf{a}$ |
| $a$ | Scalar |
| $\mathcal{X}$ | input space, i.e. the space of all possible inputs, data examples |
| $\mathcal{Y}$ | target space i.e. the space of all possible property values |
| $p(\mathbf{x}, y)$ | joint data distribution |
| $p_\theta(\mathbf{x}, y)$ | joint model parametrized by parameters $\theta \in \Theta$ |
| $p_\theta(y \mid \mathbf{x})$ | predictive model parametrized by parameters $\theta \in \Theta$ |
| $p_\theta(\mathbf{x})$ | generative model parametrized by parameters $\theta \in \Theta$ |

# D Proofs

## D.1 Gradient Interference

**Lemma D.1.** *Let $\mathbf{x} \in \mathbb{R}^I$ and define*

$$a_i = \text{softmax}(\mathbf{x})_i = \frac{\exp x_i}{\sum_{k=1}^I \exp x_k} \text{ , for } i = 1, \ldots, I.$$

*The Jacobian of the softmax is given by*

$$\frac{\partial a_i}{\partial x_j} = a_i(\delta_{ij} - a_j), \qquad i, j = 1, \ldots, I,$$

*where $\delta_{ij}$ is the Kronecker delta, i.e., $\delta_{ij} = 1$ if $i = j$ and $0$ otherwise.*

*Proof.* Differentiate the quotient $a_i = \exp x_i / \sum_k \exp x_k$ using the product and chain rules (Petersen et al., 2008). $\square$

**Corollary D.2.** *Let $\mathbf{Q}, \mathbf{K} \in \mathbb{R}^{T \times d}$ and the attention score matrix $\mathbf{S}_\rightarrow$ with a causal mask $\mathbf{M}_\rightarrow$ be defined as*

$$\mathbf{S}_\rightarrow = \frac{\mathbf{Q}\,\mathbf{K}^T}{\sqrt{d}} + \mathbf{M}_\rightarrow \text{ , where } (\mathbf{M}_\rightarrow)_{ij} = \begin{cases} 0 & , \text{ if } i \geq j \\ -\infty & , \text{ if } i < j. \end{cases}$$

*For a fixed row index $t \in [T]$, define the attention score row-vector $\mathbf{s}_t = (\mathbf{S})_t \in \mathbb{R}^T$ and the corresponding row-wise softmax output as $\mathbf{a}_t = \text{softmax}(\mathbf{s}_t) \in \mathbb{R}^T$. The Jacobian of the softmax output $\mathbf{a}_t$ with respect to masked attention score $\mathbf{s}_t$ is given by*

$$\frac{\partial(\mathbf{a}_t)_i}{\partial(\mathbf{s}_t)_j} = (\mathbf{a}_t)_i(\delta_{ij} - (\mathbf{a}_t)_j).$$

*Hence, if $i < t$ or $j < t$, while $i \neq j$, then $\frac{\partial(\mathbf{a}_t)_i}{\partial(\mathbf{s}_t)_j} = 0$.*

*Proof.* Lemma D.1 gives the derivative of the softmax. As the causal mask sets $(\mathbf{s}_t)_j = -\infty$ for every $j < t$, the corresponding probabilities satisfy $(\mathbf{a}_t)_j = 0$. $\square$

# E Benchmark Task Definitions

## E.1 Conditional Molecule Generation

**Quantitative Estimate of Drug-likeness (QED).** A continuous metric of the drug-likeness of a molecule based on physicochemical properties such as molecular weight and hydrophobicity, with values ranging from 0 to 1. (Bickerton et al., 2012)

**Synthetic Accessibility (SA).** A continuous metric quantifying how difficult a molecule is to synthesize, derived from structural complexity, where lower values indicate easier synthesis. (Ertl & Schuffenhauer, 2009)

**Partition Coefficient (logP).** A continuous metric of molecular hydrophobicity, defined as the logarithm of the partition coefficient between octanol and water, where higher values denote greater affinity for lipophilic environments. (Wildman & Crippen, 1999)

Metric values calculated using rdkit 2023.09.2.

### E.2 Out-of-Distribution Molecular Property Prediction

**DRD2-Hi.** Binary classification dataset of 8482 compounds with labels indicating dopamine receptor inhibition, with therapeutic relevance in schizophrenia and Parkinson's disease; dataset obtained from ChEMBL30. (Mendez et al., 2019)

**HIV-Hi.** Binary classification dataset of 41127 compounds from the Drug Therapeutics Program AIDS Antiviral Screen, with labels indicating the inhibition of HIV replication; dataset obtained from MoleculeNet. (Wu et al., 2018)

**KDR-Hi.** Binary classification dataset with labels indicating VEGFR2 (vascular endothelial growth factor receptor 2) inhibition, a kinase target in cancer therapy, with training restricted to 500 compounds to simulate low-data regimes; dataset obtained from Chembl30. (Mendez et al., 2019)

**Sol-Hi.** Binary classification dataset of 2173 compounds with labels indicating solubility; dataset obtained at Biogen. (Fang et al., 2023)

For further dataset and train/test splitting details, see (Steshin, 2023). Data accessed from `https://github.com/SteshinSS/lohi_neurips2023/tree/main/data/hi` [accessed 20.03.2023].

### E.3 Molecular Representation Learning and Property Prediction

**ESOL.** Regression dataset containing water solubility measurements for 1128 compounds.

**FreeSolv.** Regression dataset containing experimentally measured hydration free energy values in water for 642 compounds.

**Lipophilicity.** Regression dataset containing experimentally measured octanol/water distribution coefficients (logD at pH 7.4), a key indicator of membrane permeability and solubility, for 4,200 compounds.

**BACE.** Binary classification dataset of 1513 compounds with experimentally determined qualitative binding results for a set of inhibitors of human $\beta$-secretase 1 (BACE-1).

**BBBP.** Binary classification dataset of 2039 compounds with binary labels indicating blood–brain barrier permeability.

**ClinTox.** Multitask classification dataset of 1478 compounds with labels indicating whether a compound is (i) FDA-approved and/or (ii) failed clinical trials due to toxicity reasons.

**HIV.** Binary classification dataset of 41127 compounds from the Drug Therapeutics Program AIDS Antiviral Screen, measuring inhibition of HIV replication.

**Tox21.** Multitask classification dataset of 7831 compounds with qualitative toxicity measurements across 12 biological targets, including nuclear receptors and stress response pathways.

**ToxCast.** Multitask classification dataset of 8575 compounds with qualitative toxicity results across over 600 in vitro assays, derived from high-throughput screening.

**SIDER.** Multitask classification dataset of 1427 approved drugs, with side effects grouped into 27 system organ classes according to MedDRA classifications, capturing adverse drug reactions across organ systems.

For further details, see Table 1 in Wu et al. (2018). To ensure comparability with Uni-Mol (Zhou et al., 2023), we accessed data from `https://bioos-hermite-beijing.tos-cn-beijing.volces.com/unimol_data/finetune/molecular_property_prediction.tar.gz` [accessed 20.03.2023].

## F Benchmark Metric Definitions

**MAD.** Mean Absolute Deviation between predicted and target property values; lower is better.

**SD.** Standard Deviation of generated property values from the target; lower is better.

**Validity.** Fraction of syntactically valid molecules generated by the model; higher is better.

**Uniqueness.** Fraction of unique molecules among generated samples; higher is better.

**Novelty.** Fraction of generated molecules not present in the training set; higher is better.

**KL Div. Score.** Score based on the Kullback–Leibler Divergence between various descriptor distributions of generated and training molecules; values normalized in the range $[0, 1]$; higher values indicate a closer match between descriptor distributions between generated and training molecules. (Brown et al., 2019)

**FCD Score.** Score based on the Fréchet ChemNet Distance between the generated and reference (training) molecule embedding distributions, calculated in ChemNet feature space; values normalized in the range $[0, 1]$; higher values indicate closer resemblance of the generated to reference molecules. (Brown et al., 2019)

**Perplexity.** Exponentiated negative log-likelihood of a sequence, with the log-likelihood being calculated per token, using ProGen2-medium (Torres et al., 2025); lower values indicate greater model-based plausibility of the generated peptides.

**Diversity.** Average pairwise Levenshtein distance between the generated sequences; higher values indicate greater diversity of the generated samples. For details, see Eq. 6 in Kim et al. (2021), where Hyformer replaces Soergel with Levenshtein distance.

**Fitness.** A measure quantifying to what extent a peptide forms a stable, amphipathic $\alpha$-helix, computed according. (Li et al., 2024)

**HydrAMP MIC.** The probability of a peptide being active against E.Coli bacteria strain predicted with HydrAMP. (Szymczak et al., 2023)

**AMPlify.** The probability of a peptide being antimicrobial predicted with AMPlify. (Li et al., 2022)

**amPEPy.** The probability of a peptide being antimicrobial predicted with amPEPy. (Lawrence et al., 2020)

## G   Pre-training Details

We implement HYFORMER using a LLAMA backbone (Touvron et al., 2023). Depending on the size of the pretraining dataset, we scale HYFORMER to 8.7M parameters for GuacaMol[3] and 50M parameters for the UniMol[4] and peptide datasets [5]. These configurations align model capacity with dataset size and ensure a fair comparison with prior work: the 8.7M model is comparable to MolGPT (Bagal et al., 2022), while the 50M variant matches the scale of Uni-Mol (Zhou et al., 2023) and Graph2Seq (Gao et al., 2024b). For GuacaMol, we apply 2× data augmentation using non-canonical SMILES enumeration (Bjerrum, 2017; Arús-Pous et al., 2019) to increase molecular diversity. All models are pretrained using pre-computed molecular descriptors (Yang et al., 2019). The balancing of the tasks $(p_{\texttt{[LM]}}, p_{\texttt{[MLM]}}, p_{\texttt{[PRED]}})$ is set to $(0.90, 0.05, 0.05)$ and $(0.80, 0.10, 0.10)$, respectively.

We use SMILES (Weininger, 1988) or amino acid sequences as molecular representations across all experiments. For tokenization, we adopt an extended character-level tokenizer for SMILES, based on Schwaller et al. (2020), and use the ESM-2 tokenizer (Lin et al., 2023) for peptides.

We pre-train HYFORMER using a batch size of 1024 for up to 50K or 250K iterations, depending on model size. Training is performed with the AdamW optimizer ($\beta_1 = 0.9$, $\beta_2 = 0.95$, $\epsilon = 1 \times 10^{-5}$, weight decay $= 1 \times 10^{-1}$), using a peak learning rate of $6 \times 10^{-4}$ with cosine decay and 5000 warm-up steps. We use gradient clipping with a maximum norm of 1.0. All input sequences are padded to a fixed length of 128 tokens. Training is conducted using bfloat16 precision on a single NVIDIA H100 80GB HBM3 GPU.

---

[3]Data accessed from `https://figshare.com/projects/GuacaMol/56639` on 20.03.2025.

[4]Data accessed from `https://bioos-hermite-beijing.tos-cn-beijing.volces.com/unimol_data/finetune/molecular_property_prediction.tar.gz` on 20.03.2023.

[5]Data accessed from `https://app.peptipedia.cl/`, `https://www.uniprot.org/uniprotkb?query=%28length`, `https://ampsphere.big-data-biology.org/downloads` and `https://drive.google.com/drive/folders/1krim1ugqNDmgmHZCFSOvmynWxCSzyOto` on 17.04.2025 with train/test set constructed using standard scikit's train/test splitting and random seed 44.

Table 7: Architectural details of HYFORMER.

| NUM. PARAM. | EMBED. DIM | HIDDEN DIM | #LAYERS | # ATT. HEADS |
|---|---|---|---|---|
| 8.7M | 256 | 1024 | 8 | 8 |
| 50M | 512 | 2048 | 12 | 8 |

# H   Experimental Details

All fine-tuning and inference is conducted using float32 precision on a single NVIDIA V100 32GB GPU.

## H.1   Conditional Molecule Generation

We jointly fine-tune HYFORMER, pretrained on GuacaMol dataset, for 10 epochs with a batch size of 256. The peak learning rate is selected from the set $\{1e-4, 2e-4, 3e-4, 4e-4, 5e-4, 5e-4, 6e-4\}$, based on root mean squared error (RMSE) with respect to the target property. During fine-tuning, we set the task probability vector to $(p_{\texttt{[LM]}}, p_{\texttt{[PRED]}}) = (0.5, 0.5)$ and do not perform hyperparameter search over this setting, as it yields satisfactory performance by default. For the non-joint variant of HYFORMER, we freeze the pretrained model and fine-tune only the prediction head. This avoids catastrophic forgetting of the generative capability when removing the generative loss during training. For each target property value, we sample 100K unique molecules, with a wall-clock time of $78 \pm 1$ seconds, and retain those passing a manually defined threshold, using multinomial top-$k$ sampling with $\tau = 0.9$ and $k = 10$. Note that reported SA scores are normalized, following (Gao et al., 2024b).

To further characterize the selectivity of conditional sampling and the calibration of the predictive heads, we report acceptance rates in the conditional molecule generation experiment in Table 9. The results confirm that conditional sampling with HYFORMER is highly selective across all target values.

Table 8: Conditional generative performance on GuacaMol dataset across all targets. Best model is marked **bold**.

| | PRETRAIN | JOINT | METRIC | QED=0.5 | QED=0.7 | QED=0.9 | SA=0.7 | SA=0.8 | SA=0.9 | LOGP=0.0 | LOGP=2.0 | LOGP=4.0 | AVG. |
|---|---|---|---|---|---|---|---|---|---|---|---|---|---|
| MolGPT | ✗ | ✗ | MAD ↓ | 0.081 | 0.082 | 0.097 | 0.024 | 0.019 | 0.013 | 0.304 | 0.239 | 0.286 | 0.127 |
| | | | SD ↓ | 0.065 | 0.066 | 0.092 | 0.022 | 0.016 | 0.013 | 0.295 | 0.232 | 0.258 | 0.118 |
| | | | VALIDITY ↑ | 0.985 | 0.985 | 0.984 | 0.975 | 0.988 | 0.995 | 0.982 | 0.983 | 0.982 | 0.984 |
| GraphGPT-1W-C | ✗ | ✗ | MAD ↓ | 0.041 | 0.031 | 0.077 | 0.012 | 0.028 | 0.031 | 0.103 | 0.189 | 0.201 | 0.079 |
| | | | SD ↓ | 0.079 | 0.077 | 0.121 | 0.055 | 0.062 | 0.070 | 0.460 | 0.656 | 0.485 | 0.229 |
| | | | VALIDITY ↑ | 0.988 | 0.995 | 0.991 | 0.995 | 0.991 | 0.998 | 0.980 | **0.992** | 0.991 | 0.991 |
| | ✓ | ✗ | MAD ↓ | 0.032 | 0.033 | 0.051 | **0.002** | 0.009 | 0.022 | **0.017** | 0.190 | 0.268 | 0.069 |
| | | | SD ↓ | 0.080 | 0.075 | 0.090 | 0.042 | 0.037 | 0.062 | 0.463 | 0.701 | 0.796 | 0.261 |
| | | | VALIDITY ↑ | **0.996** | **0.998** | **0.999** | **0.995** | **0.999** | 0.996 | 0.994 | 0.990 | 0.992 | **0.995** |
| HYFORMER | ✓ | ✗ | MAD ↓ | 0.035 (0.000) | 0.032 (0.001) | 0.027 (0.007) | 0.020 (0.001) | 0.016 (0.000) | 0.009 (0.001) | 0.131 (0.012) | 0.135 (0.007) | 0.127 (0.011) | 0.059 (0.004) |
| | | | SD ↓ | 0.049 (0.000) | 0.046 (0.002) | 0.039 (0.010) | 0.027 (0.002) | 0.021 (0.000) | 0.012 (0.001) | 0.162 (0.015) | 0.174 (0.010) | 0.175 (0.016) | 0.078 (0.006) |
| | | | VALIDITY ↑ | 0.993 (0.003) | 0.993 (0.003) | 0.993 (0.004) | 0.986 (0.009) | 0.985 (0.001) | **0.999 (0.002)** | 0.978 (0.031) | 0.983 (0.004) | **0.995 (0.007)** | 0.989 (0.007) |
| | ✓ | ✓ | MAD ↓ | **0.010 (0.001)** | **0.009 (0.000)** | **0.006 (0.000)** | 0.008 (0.001) | **0.005 (0.000)** | **0.001 (0.000)** | 0.033 (0.005) | **0.044 (0.001)** | **0.046 (0.001)** | **0.018 (0.001)** |
| | | | SD ↓ | **0.018 (0.002)** | **0.018 (0.002)** | **0.010 (0.003)** | **0.015 (0.003)** | **0.009 (0.002)** | **0.004 (0.000)** | **0.037 (0.007)** | **0.057 (0.003)** | **0.059 (0.001)** | **0.025 (0.003)** |
| | | | VALIDITY ↑ | 0.983 (0.009) | 0.990 (0.006) | 0.996 (0.005) | 0.976 (0.005) | 0.981 (0.002) | **0.999 (0.002)** | **1.000 (0.000)** | 0.991 (0.007) | 0.971 (0.011) | 0.987 (0.005) |

---

**Algorithm 2** Conditional sampling with HYFORMER

**Input:** Number of examples to sample $K$, batch size $B$, condition $Y$, model parameters $\theta$.

1: $\mathcal{D}_{sampled} = \emptyset$
2: **while** $|\mathcal{D}_{sampled}| < K$ **do**
3:     Sample $B$ many examples $(\mathbf{x}, y) \sim p_\theta(\mathbf{x}, y)$
4:     Accept examples $\mathcal{D}_{batch} = \{(\mathbf{x}, y) \mid y \in Y\}$
5:     Append dataset $\mathcal{D}_{sampled} = \mathcal{D}_{sampled} \cup \mathcal{D}_{batch}$
6: **end while**

---

Table 9: Number of accepted samples per 100,000 generated molecules in the conditional generation experiment. Mean and standard deviation across three random seeds. The average (Avg.) is computed over all target values.

| Model | QED=0.5 | QED=0.7 | QED=0.9 | SA=0.7 | SA=0.8 | SA=0.9 | logP=0.0 | logP=2.0 | logP=4.0 | Avg. |
|---|---|---|---|---|---|---|---|---|---|---|
| HYFORMER (no-joint) | 315 (7) | 367 (11) | 162 (11) | 144 (5) | 448 (15) | 229 (23) | 14 (3) | 76 (7) | 70 (2) | 203 (81) |
| HYFORMER (joint) | 295 (19) | 330 (4) | 176 (17) | 140 (10) | 426 (13) | 254 (7) | 11 (4) | 70 (7) | 67 (4) | 197 (71) |

## H.2 Out-of-Distribution Molecular Property Prediction Task

We use HYFORMER pre-trained on UniMol dataset and perform a grid search over hyperparameters, as detailed in Table 10, with end-to-end joint fine-tuning, with early stopping triggered if the validation loss does not improve for 5 consecutive epochs. Results in Table 2 are reported from (Steshin, 2023).

Table 10: Hyperparameter ranges for the grid search hyperparameter optimization on out-of-distribution molecular property prediction task.

| HYPERPARAMETER | SEARCH RANGE |
|---|---|
| MAX EPOCHS | {20, 50, 100} |
| BATCH SIZE | {64, 128, 256} |
| LEARNING RATE | [1E-5, 6E-4] |
| WEIGHT DECAY | [1E-2, 1E-1] |
| POOLER DROPOUT | [0.0, 0.2] |
| LEARNING RATE DECAY | {TRUE, FALSE} |
| $(p_{[\text{LM}]}, p_{[\text{PRED}]})$ | {(0.0, 1.0), (0.1, 0.9)} |

## H.3 Molecular Representation Learning Task

For KNN probe, we use the Euclidean norm to pick K most similar molecules. For each dataset, we search the parameter K in the set $\{1, 3, 5, 100, 300, 500, 1000, 3000, 5000\}$ and pick K with the best performance on the validation split. For linear probe, we report the results of linear probe with L2 regularization added. If the validation loss between the epochs does not decrease by more than 0.0001 for 10 consecutive epochs, we terminate the training process early. All results in Table 3 are ours.

## H.4 Molecule Generation Task

For generation, we use HYFORMER pre-trained on GuacaMol and sample using multinomial top-$k$ sampling, with $k = 10$ and varying temperature $\tau = \{0.9, 1.0, 1.1\}$.

In Table 4, baseline results for JTVAE and MAGNeT are reported from (Hetzel et al., 2023), for MoLeR and MiCaM from (Geng et al., 2023), for VAE, LSTM from (Brown et al., 2019), for MolGPT from (Bagal et al., 2022).

## H.5 Molecular Property Prediction Task

We use HYFORMER pre-trained on UniMol dataset and perform a grid search over hyperparameters, as detailed in Table 11, with end-to-end predictive fine-tuning run for a maximum of 20 epochs, with early stopping triggered if the validation loss does not improve for 5 consecutive epochs. Results in Table 5 are reported from (Zhou et al., 2023; Gao et al., 2024b).

## H.6 Antimicrobial Peptide Design

**Dataset** We construct a general-purpose peptide dataset and an AMP-specific dataset. For the general purpose dataset, we collect 3459247 peptide sequences with length 8-50 from the combined Peptipedia (Cabas-Mora et al., 2024) and UniProt (Consortium, 2024) datasets and apply CDHIT filtering with a similarity threshold of 90%. For the AMP-specific dataset, we collect 1056321 sequences from combining the Peptipedia (Cabas-Mora et al., 2024), filtered with Antigram (-), Antigram (+), Antibacterial and

Table 11: Hyperparameter ranges for the grid search hyperparameter optimization on molecular property prediction task.

| HYPERPARAMETER | SEARCH RANGE |
|---|---|
| BATCH SIZE | {16, 64, 128, 256} |
| LEARNING RATE | [1E-5, 1E-3] |
| WEIGHT DECAY | [1E-2, 3E-1] |
| POOLER DROPOUT | [0.0, 0.2] |
| LEARNING RATE DECAY | {TRUE, FALSE} |

Table 12: Unconditional generative performance on MOSES benchmark. The best model in each category is marked **bold**.

| MODEL | VALIDITY ↑ | UNIQUE ↑ | NOVELTY ↑ | INTDIV1 ↑ | INTDIV2 ↑ |
|---|---|---|---|---|---|
| *UNCONDITIONAL* | | | | | |
| HMM | 0.076 | 0.567 | **0.999** | 0.847 | 0.810 |
| NGRAM | 0.238 | 0.922 | 0.969 | **0.874** | 0.864 |
| COMBINATORIAL | **1.000** | 0.991 | 0.988 | 0.873 | **0.867** |
| CHARRNN | 0.975 | 0.999 | 0.842 | 0.856 | 0.850 |
| VAE | 0.977 | 0.998 | 0.695 | 0.856 | 0.850 |
| AEE | 0.937 | 0.997 | 0.793 | 0.856 | 0.850 |
| LATENTGAN | 0.897 | 0.997 | 0.949 | 0.857 | 0.850 |
| JT-VAE | **1.000** | 0.999 | 0.914 | 0.855 | 0.849 |
| MOLGPT | 0.994 | **1.000** | 0.797 | 0.857 | 0.851 |
| HYFORMER$_{\tau=0.9}$ | 0.996 | **1.000** | 0.701 | 0.851 | 0.845 |
| HYFORMER$_{\tau=1.0}$ | 0.991 | **1.000** | 0.749 | 0.856 | 0.850 |
| HYFORMER$_{\tau=1.1}$ | 0.986 | **1.000** | 0.791 | 0.861 | 0.855 |
| *FEW-SHOT* | | | | | |
| GRAPHGPT-1W$_{s=0.25}$ | **0.995** | 0.995 | 0.255 | 0.854 | 0.850 |
| GRAPHGPT-1W$_{s=0.5}$ | 0.993 | 0.996 | 0.334 | 0.856 | 0.848 |
| GRAPHGPT-1W$_{s=1.0}$ | 0.978 | 0.997 | 0.871 | **0.860** | **0.857** |
| GRAPHGPT-1W$_{s=2.0}$ | 0.972 | **1.000** | **1.000** | 0.850 | 0.847 |

Antimicrobial keywords, Uniprot with the keywords antimicrobial and AMPSphere (Santos-Júnior et al., 2022), and applying CDHIT filtering with a similarity threshold of 90%.

**Pre-trainig** We pre-train HYFORMER in a two-stage manner, by first training on the general-purpose, followed by training on the AMP specific dataset with peak learning rate equal to 4e−4. All additional details follow Appendix G.

**Fine-tuning** We fine-tune HYFORMER for a maximum of 10 epochs, with batch size 64, peak learning rate 5e−5 and early stopping, with task probabilities $(p_{\texttt{[LM]}}, p_{\texttt{[PRED]}})$ equal to (0.6, 0.4). Additionally, we freeze the first four layers of the model.

**Conditional Sampling** For the antimicrobial peptide design experiment, we unconditionally sample around 6.7M sequences and accept 50K of them, which results in an acceptance rate of around 0.7%. Note that our sampling procedure intentionally prioritizes high selectivity over throughput. All peptides have a maximum length of 50 AAs.

# I Additional Experiments

## I.1 Unconditional Molecule Generation on MOSES benchmark

To additionally evaluate the unconditional generative performance of HYFORMER, we perform an evaluation on the MOSES benchmark. Analogously to unconditional molecule generation in Section 5.2.1, we scale HYFORMER to 8.5M parameters and follow all the training details in Appendix G for GuacaMol dataset. We compare HYFORMER, across various sampling temperatures $\tau$, to baseline unconditional and few-shot generative models, as reported in (Gao et al., 2024b).

HYFORMER successfully generates valid, unique, novel and diverse molecules, rivaling other unconditional and few-shot generative models.

### I.2   Qualitative Evaluation of Generated Molecules

To investigate the effect of sampling temperature on the structural diversity and chemical quality of generated molecules, we show molecules sampled in the unconditional generation task (Section 5.2.1), at temperatures $\tau = 0.9$, 1.0, and 1.1. For each sampled molecule, we additionally report four chemical properties: molecular partition coefficient (LogP), topological polar surface area (TPSA), quantitative estimate of drug-likeness (QED) and molecular weight (MW). At $\tau = 0.9$, the model generates drug-like molecules, with the majority exhibiting QED $\geq 0.7$ and MW $< 500$ g/mol (Fig. 3). At $\tau = 1.0$, the sampling process yields molecules with greater structural diversity (Fig. 4). Despite the increased exploration of chemical space, some molecules exhibit lower QED values. At $\tau = 1.1$, the model produces molecules with less common substituent patterns. Some of these structures exceed traditional drug-likeness thresholds, such as MW $> 500$ g/mol or LogP $> 5$, according to Lipinski's Rule of Five (Fig. 5). Additionally, we investigate molecules generated in the conditional generation task in Section 5.1.1 (Figure 6, 7 and 8).

### I.3   Qualitative Evaluation of Learned Representations

We next examine the Hyformer embeddings in the context of the chemical properties of the molecules (Fig. 9). To this end, we randomly sample 20,000 molecules and pass them through HYFORMER's encoder, pre-trained for molecular property prediction in Section 5.2.2, to obtain molecule embeddings. We visualize the embeddings in two dimensions through principal components analysis (PCA) and color them according to their four chosen chemical properties (LogP, TPSA, QES, MW).

Qualitatively, the spatial arrangement of molecules is clearly connected to their chemical properties. Furthermore, embeddings exhibit a smooth profile of change w.r.t. each property. These observations indicate that HYFORMER learns well-behaved, information-rich molecular representations.

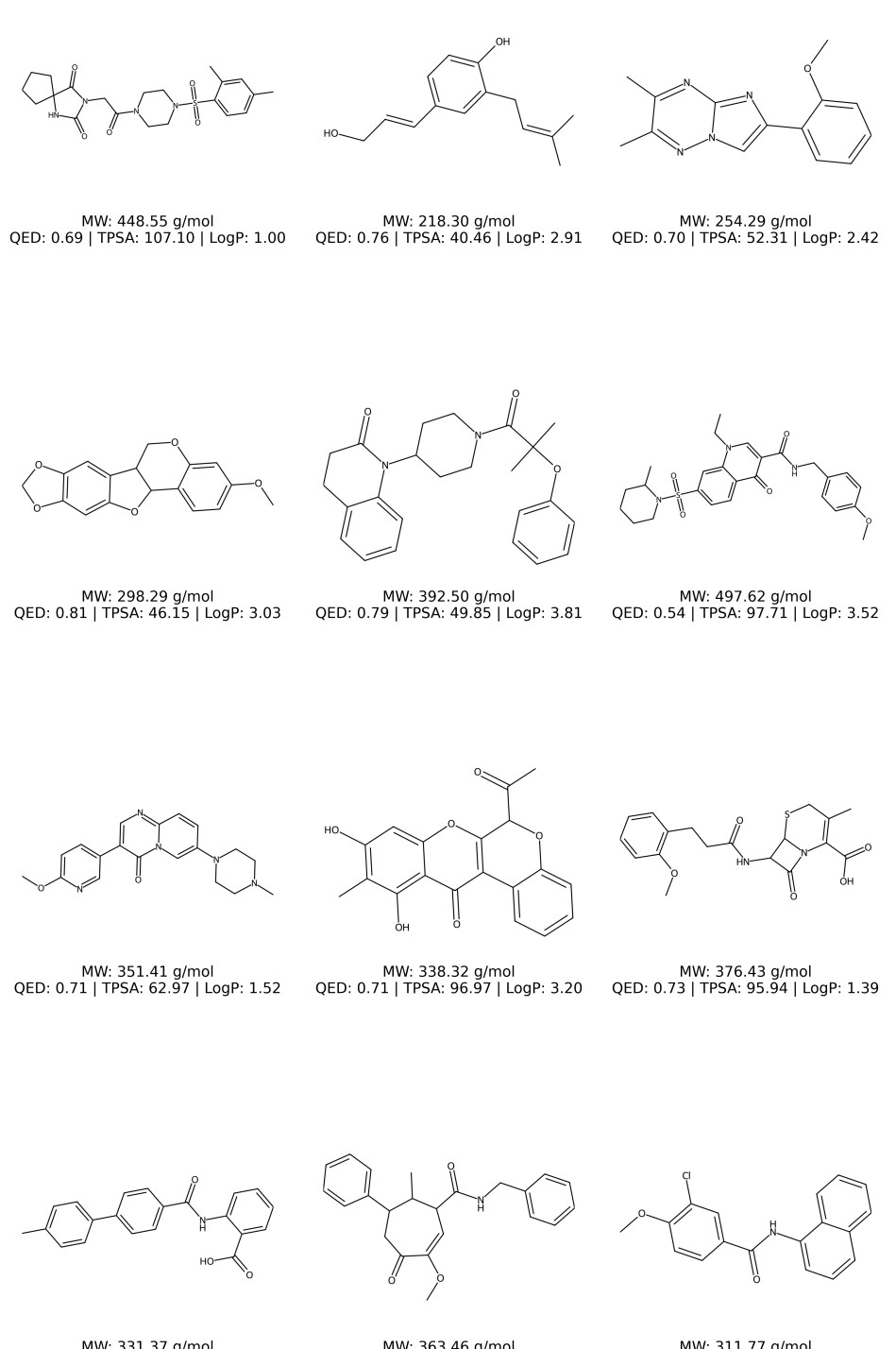

Figure 3: Structures of the twelve generated molecules with Hyformer when the sampling temperature is 0.9, visualized using RDKit, together with their properties.

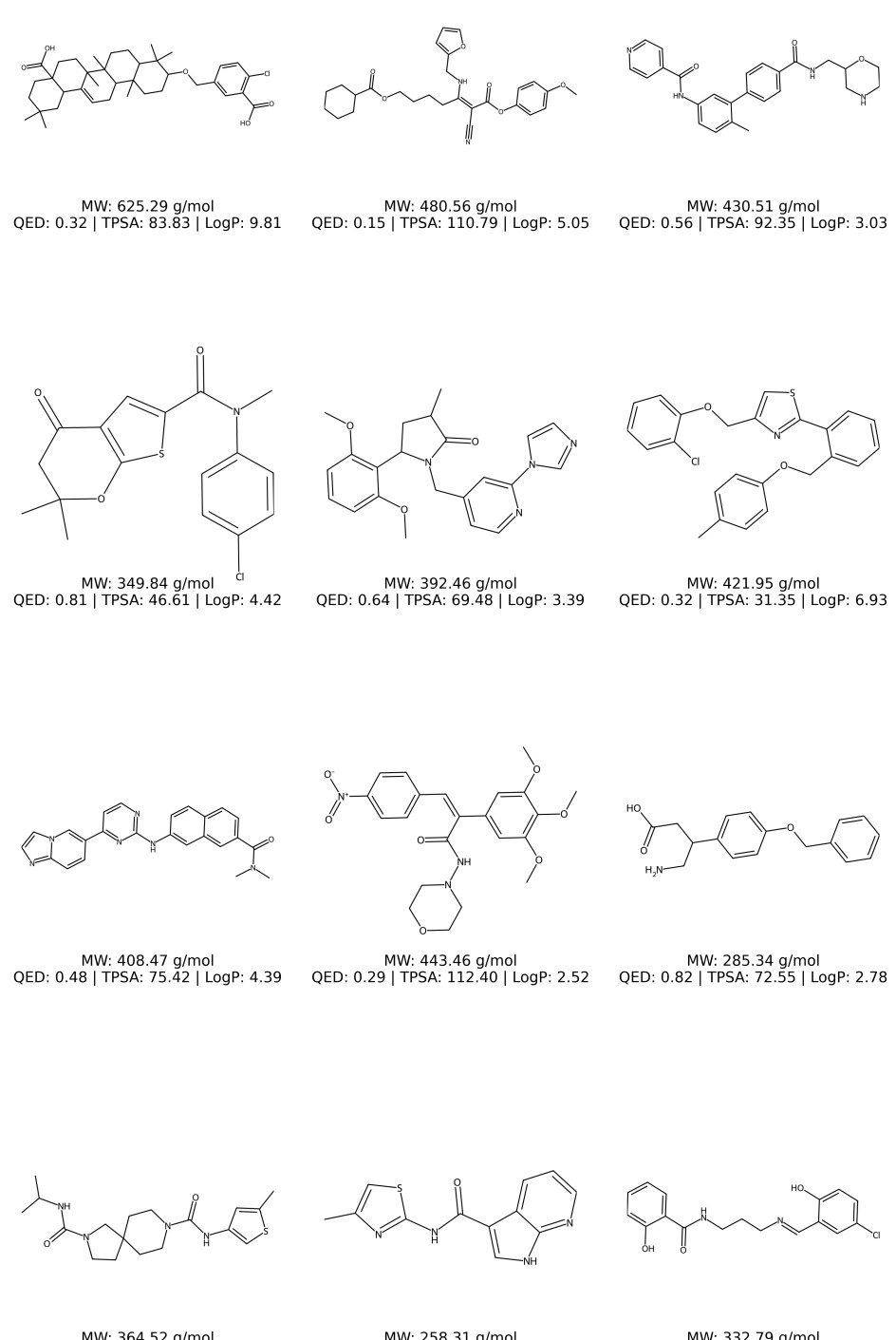

Figure 4: Structures of the twelve generated molecules with Hyformer when the sampling temperature is 1.0, visualized using RDKit, together with their properties.

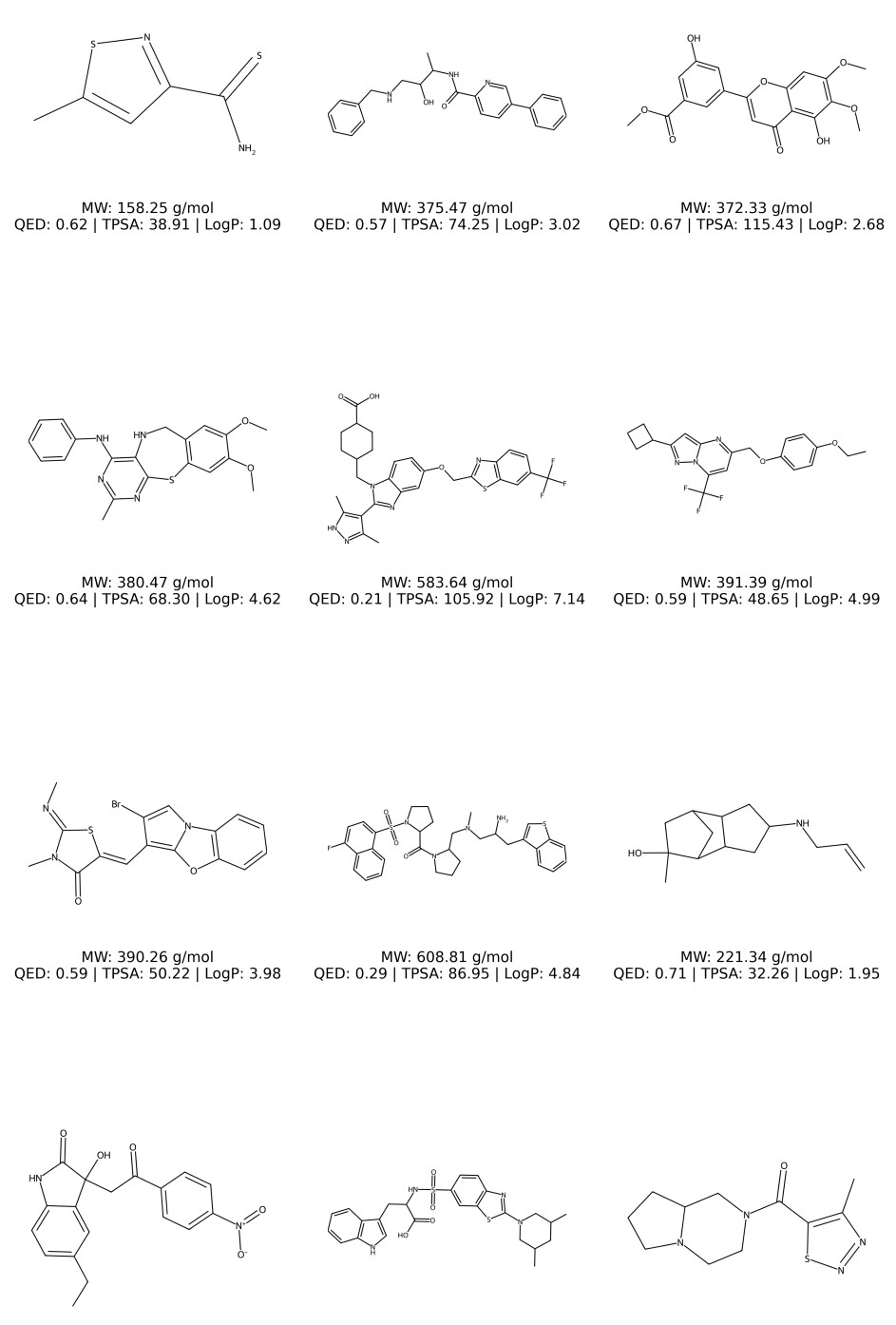

Figure 5: Structures of the twelve generated molecules with Hyformer when the sampling temperature is 1.1, visualized using RDKit, together with their properties.

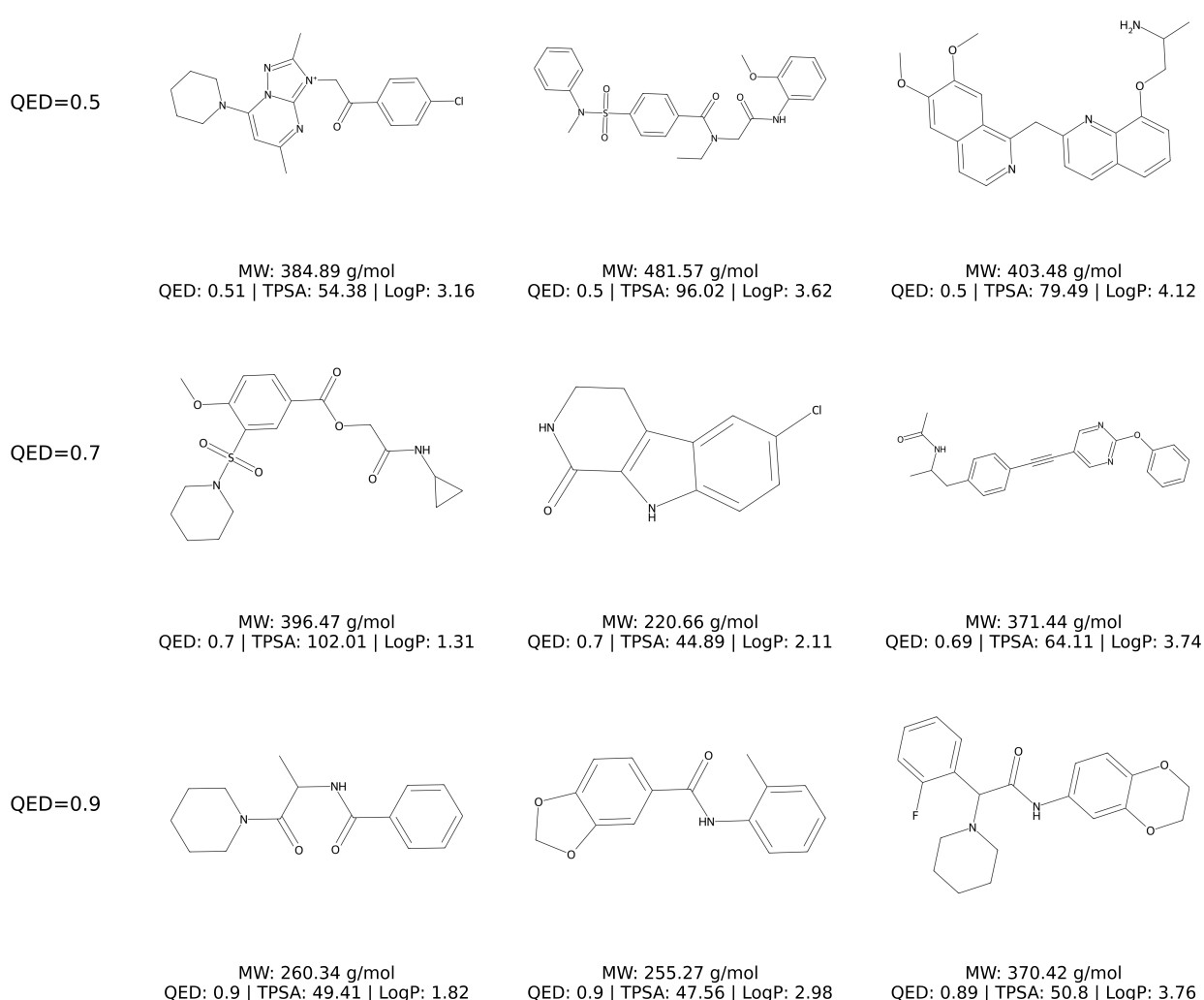

QED=0.5

MW: 384.89 g/mol
QED: 0.51 | TPSA: 54.38 | LogP: 3.16

MW: 481.57 g/mol
QED: 0.5 | TPSA: 96.02 | LogP: 3.62

MW: 403.48 g/mol
QED: 0.5 | TPSA: 79.49 | LogP: 4.12

QED=0.7

MW: 396.47 g/mol
QED: 0.7 | TPSA: 102.01 | LogP: 1.31

MW: 220.66 g/mol
QED: 0.7 | TPSA: 44.89 | LogP: 2.11

MW: 371.44 g/mol
QED: 0.69 | TPSA: 64.11 | LogP: 3.74

QED=0.9

MW: 260.34 g/mol
QED: 0.9 | TPSA: 49.41 | LogP: 1.82

MW: 255.27 g/mol
QED: 0.9 | TPSA: 47.56 | LogP: 2.98

MW: 370.42 g/mol
QED: 0.89 | TPSA: 50.8 | LogP: 3.76

Figure 6: Structures of molecules generated by Hyformer conditioned on QED values, visualized using RDKit, along with their chemical properties.

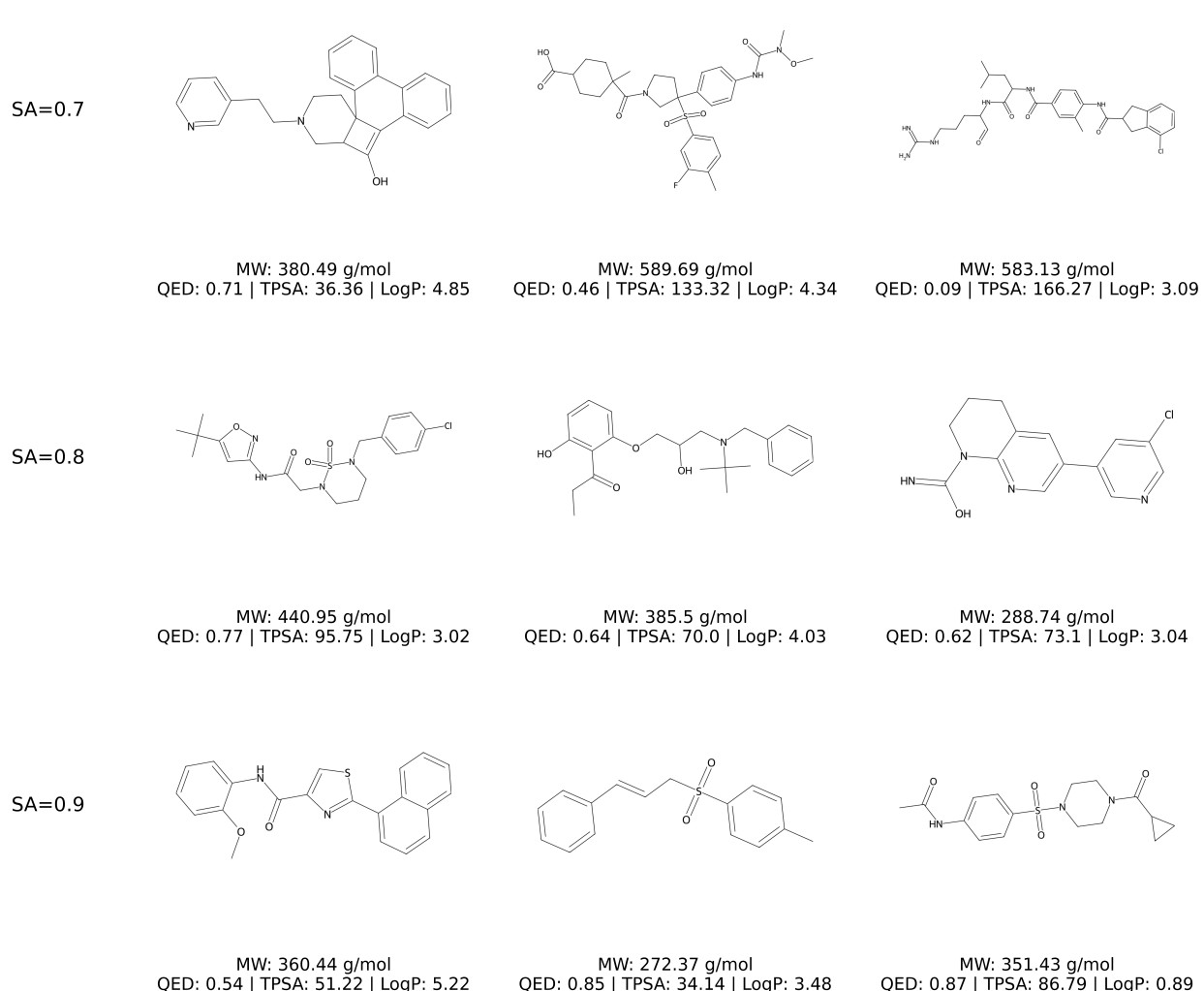

Figure 7: Structures of molecules generated by Hyformer conditioned on SA score, visualized using RDKit, along with their chemical properties.

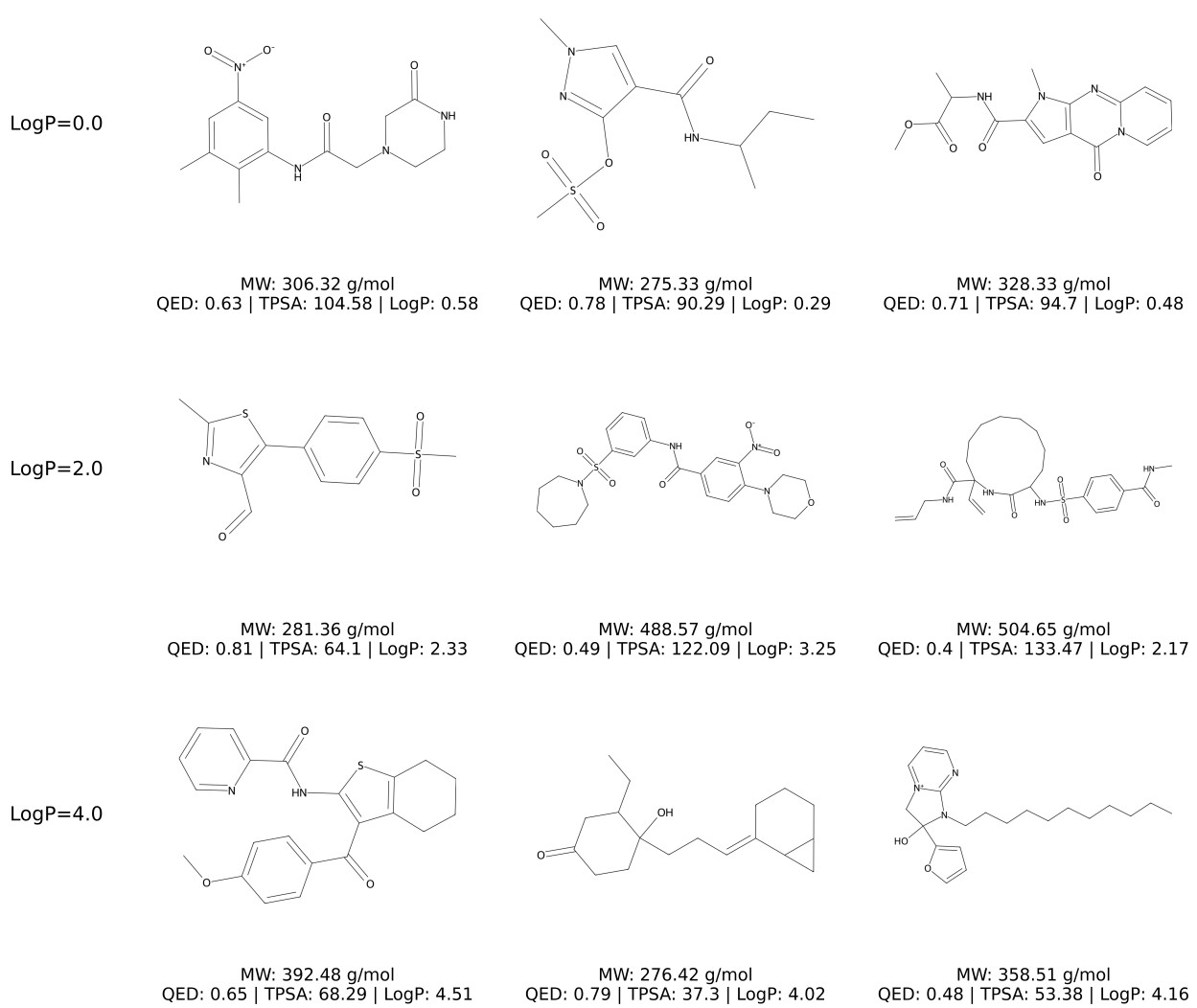

Figure 8: Structures of molecules generated by Hyformer conditioned on LogP values, visualized using RDKit, along with their chemical properties.

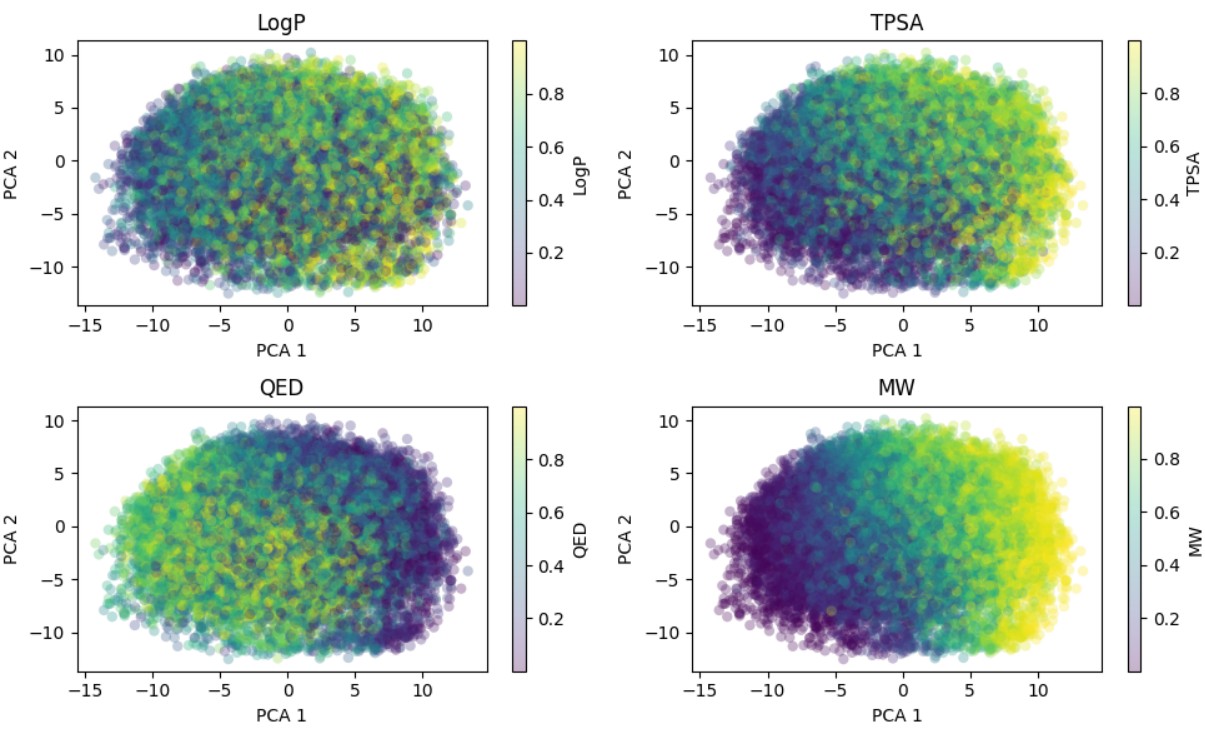

Figure 9: Hyformer's molecular embeddings. The considered chemical properties are normalized to lie in the $[0, 1]$ interval.

