# OpenReview forum: "Synergistic Benefits of Joint Molecule Generation and Property Prediction"
_TMLR — Accepted by TMLR_

### Review · Reviewer_xgqr · 2025-09-15

**Summary Of Contributions:**

The paper tackles joint generative and predictive modelling for small molecular structures. The proposed Hyformer is based on Transformer encoder for prediction with molecule string inputs, and jointly with Transformer decoder for generative modelling of molecular strings. The joint approach yields strong results on both generative and predictive molecular benchmarks.

**Additional Comments:**

I would be interested in scaling studies where Hyformer is trained on very large scale molecular datasets annotated with properties. Can the model zero-shot generalize to all other molecular benchmarks in that case? If yes, that would be extremely exciting.

Additionally, currently Hyformer needs Transformer encoder layers. What if these are removed, and property prediction is also framed as a generative task? If that works, it would also be extremely exciting.

Lastly, how would Hyformer generalize to considering molecular interactions? E.g. small molecules + protein partners.

**Audience:**

Yes

**Audience Explanation:**

Yes, I believe there is now a significant community of machine learners interested in molecular modelling applications, who will find this work exciting and forward looking for its unified perspective on prediction + generation.

**Broader Impact Concerns:**

No concerns.

**Claims And Evidence:**

Yes

**Claims Explanation:**

Yes, I believe that all the claims are accurate and backed up by experimental results. The experimental setup is well described in detail in the appendix, and appears to largely follow previous published papers in terms of how the benchmarks are set up, etc. At this point, molecular benchmarks are pretty standardized, so I am fairly confident that the authors implemented them correctly.

**Requested Changes:**

I realise that some past work may have ignored this, but I would suggest adding standard deviations to tables which currently miss them. As the paper claims to be the top performing model on several benchmarks, it is expected that future work will be asked to compare to Hyformer. It would be good to know how much variation there is in the generative model's performance when varying random seed, for instance, in table 1 on conditional generation.

---

> ### Author Response · Authors · 2025-10-17
> **Additional Results to Reviewer xgqr**
>
> ## Conditional Molecule Generation (Section 5.1.1)
>
> Table 1: Conditional generative performance on GuacaMol dataset. Mean and std. dev. over 3 random seeds. Average (Avg.) is computed across targets. Best model is marked in **bold**.
>
> | Model     | Joint | Metric             | QED             | SA              | logP            | Avg.            |
> |------------|:------:|--------------------|------------------|------------------|------------------|------------------|
> | **Hyformer** | ✗ | MAD ↓        | 0.031 (0.003)   | 0.015 (0.001)   | 0.131 (0.010)   | 0.059 (0.004)   |
> |            |   | SD ↓          | 0.045 (0.004)   | 0.020 (0.001)   | 0.170 (0.014)   | 0.078 (0.006)   |
> |            |   | Validity ↑    | 0.993 (0.003)   | 0.990 (0.004)   | 0.985 (0.014)   | 0.989 (0.007)   |
> |            | ✅ | MAD ↓        | **0.008 (0.001)** | **0.005 (0.000)** | **0.041 (0.002)** | **0.018 (0.001)** |
> |            |   | SD ↓          | **0.015 (0.002)** | **0.009 (0.002)** | **0.051 (0.004)** | **0.025 (0.003)** |
> |            |   | Validity ↑    | 0.990 (0.007)   | 0.985 (0.003)   | 0.987 (0.006) | 0.987 (0.005)   |
>
>
> Table 8: Conditional generative performance on GuacaMol dataset across all targets Mean and std. dev. over 3 random seeds. Average (Avg.) is computed across targets. Best model is marked in  **bold**.
>
> | Model | Joint | Metric | QED=0.5 | QED=0.7 | QED=0.9 | SA=0.7 | SA=0.8 | SA=0.9 | logP=0.0 | logP=2.0 | logP=4.0 | Avg. |
> |:------|:------:|:--------|:--------:|:--------:|:--------:|:--------:|:--------:|:--------:|:---------:|:---------:|:---------:|:------:|
> | **Hyformer** | ✗ | MAD ↓ | 0.035 (0.000) | 0.032 (0.001) | 0.027 (0.007) | 0.020 (0.001) | 0.016 (0.000) | 0.009 (0.001) | 0.131 (0.012) | 0.135 (0.007) | 0.127 (0.011) | 0.059 (0.004) |
> |  |  | SD ↓ | 0.049 (0.000) | 0.046 (0.002) | 0.039 (0.010) | 0.027 (0.002) | 0.021 (0.000) | 0.012 (0.001) | 0.162 (0.015) | 0.174 (0.010) | 0.175 (0.016) | 0.078 (0.006) |
> |  |  | Validity ↑ | 0.993 (0.003) | 0.993 (0.003) | 0.993 (0.004) | 0.986 (0.009) | 0.985 (0.001) | **0.999 (0.002)** | 0.978 (0.031) | 0.983 (0.004) | **0.995 (0.007)** | 0.989 (0.007) |
> |  | ✅ | MAD ↓ | **0.010 (0.001)** | **0.009 (0.000)** | **0.006 (0.001)** | 0.008 (0.001) | **0.005 (0.000)** | **0.001 (0.000)** | 0.033 (0.005) | **0.044 (0.001)** | **0.046 (0.001)** | **0.018 (0.001)** |
> |  |  | SD ↓ | **0.018 (0.002)** | **0.018 (0.002)** | **0.010 (0.003)** | **0.015 (0.003)** | **0.009 (0.002)** | **0.004 (0.000)** | **0.037 (0.007)** | **0.057 (0.003)** | **0.059 (0.001)** | **0.025 (0.003)** |
> |  | | Validity ↑ | 0.983 (0.009) | 0.990 (0.006) | 0.996 (0.005) | 0.976 (0.005) | 0.981 (0.002) | **0.999 (0.002)** | **1.000 (0.000)** | 0.991 (0.007) | 0.971 (0.011) | 0.987 (0.005) |
>
> ## Unconditional Molecule Generation (Section 5.2.1)
>
> Table 3: Unconditional generative performance on GuacaMol distribution learning benchmarks. The best model in each category is marked **bold**.
>
> | Model     | FCD Score     | KL Div. Score     | Val.      | Uniq.    | Nov.      |
> |:------|:-------------|:---------------|:--------------|:--------------|:--------------|
> | Hyformer_{t=0.9}   | 0.897 (0.002) | **0.995 (0.000)** | **0.986 (0.001)** | 0.999 (0.000) | 0.879 (0.006) |
> | Hyformer_{t=1.0}   | **0.918 (0.002)** | 0.989 (0.001) | 0.978 (0.000) | 0.999 (0.000) | 0.908 (0.002) |
> | Hyformer_{t=1.1}   | 0.894 (0.002) | 0.977 (0.001) | 0.965 (0.001) | **1.000 (0.000)** | 0.931 (0.001) |
>
>
> ## Antimicrobial Peptide Design (Section 5.3)
>
> Table 6: Conditional generative performance on antimicrobial peptide design. Mean and standard deviation computed over 100 bootstrap iterations. The best model is marked **bold**.
> | Model        | Perplexity           | Diversity ↑         | Fitness ↑           | HydrAMP_MIC ↑ | AMPlify ↑ | amPEPpy ↑ |
> |:-------------|---------------------:|--------------------:|--------------------:|------------:|----------:|----------:|
> | Pepcvae | 20.11 (0.1385) | **0.87 (0.0003)** | 0.07 (0.0004) | 0.20 (0.0016) | 0.49 (0.0016) | 0.52 (0.0007) |
> | AMPgan | 18.58  (0.0966) | 0.81 (0.0005) | 0.12 (0.0005) | 0.32 (0.0019) | 0.64 (0.0018) |  0.54 (0.0008) |
> | HydrAMP | 20.14 (0.1199) | 0.86  (0.0004) | 0.09 (0.0003) | 0.49 (0.0021) | 0.59 (0.0016) | 0.52 (0.0.0006) |
> | AMPDiffusion | 16.93 (0.1776) | 0.82 (0.0004) | 0.13 (0.0005) | 0.26 (0.0018) | 0.20  (0.0014) | 0.38 (0.0006) |
> | Hyformer | 17.98 (0.0563) | 0.80 (0.0005) | **0.19 (0.0006)** | **0.80 (0.0019)** | **0.94 (0.0027)** |  **0.72 (0.0018)** |

---

> ### Author Response · Authors · 2025-10-17
> **Response to Reviewer xgqr**
>
> We thank the reviewer for the thoughtful and constructive feedback.
>
> **Standard deviations.** In the revised version, we included standard deviations over three random seeds for Hyformer on the unconditional and conditional generation benchmarks (Tables 1, 8 and 3, respectively), see Additional Results for Reviewer. For Table 6, as all methods provide a list of generated peptides, we report mean and standard deviations, calculated over 100 bootstrap iterations, for all methods.
>
> **Scaling and zero-shot generalization.** We appreciate the suggestion to explore larger-scale training. In future work, we plan to pretrain Hyformer on larger datasets to determine scaling laws. However, scaling Hyformer beyond 50M parameters and on bigger pre-training datasets would sacrifice direct comparability to existing baselines, which we aimed to preserve in this manuscript.
>
> As for the ability of Hyformer to generalize to unseen tasks, we follow the set-up of Steshin (2023) and additionally probe Hyformer on the DRD2-Hi and Sol-Hi tasks (Table 2 in Section 5.1.2) with a kNN probe. We observe promising preliminary results, where Hyformer outperforms both ECFP4 and MACCS fingerprints, indicating a strong ability of Hyformer to generalize to downstream prediction tasks. In the final version of the manuscript, we additionally plan to probe Hyformer with GB, SVM and MLP probe across all tasks, to adhere to Table 2 in Section 5.1.2 and Steshin (2023).
>
> | Model             | DRD2-Hi | Sol-Hi |
> |:------------------|:--------------------:|:-------------------:|
> | kNN (Hyformer)    | 0.721 (0.038)        | 0.484 (0.006)       |
>
>
> **Predictive-only via generative framing.** We agree that framing property prediction as a purely generative task is an interesting and promising direction. We will mention this as part of future work in the revised version. However, here an additional obstacle is how to tokenize continuous molecular properties.
>
> **Molecular interactions.** We appreciate this insightful question. Since Hyformer natively supports multimodal inputs, extending it to model small-molecule–protein interactions is straightforward. We are currently exploring this by conditioning the decoder on protein embeddings, and plan to combine this with additional pretraining objectives to unify different molecular modalities. This consideration will be added to the Discussion section.
>
> We thank the reviewer again for the constructive comments, which will help us strengthen the final version of the paper.

---

> > ### Comment · Reviewer_xgqr · 2025-10-19
> >
> > The author’s rebuttal addresses my main concerns.

---

### Review · Reviewer_fGqQ · 2025-09-23

**Summary Of Contributions:**

The authors propose Hyformer, a transformer-based joint model for molecular generation and property prediction. Unlike prior approaches that train separate models, Hyformer shares a single backbone across both tasks and switches between autoregressive (causal) and bidirectional attention modes via task tokens. Two specialized heads handle generative and predictive outputs. The model is evaluated on standard molecular ML benchmarks (e.g., GuacaMol, MoleculeNet, Lo-Hi) and demonstrates strong performance in unconditional/conditional generation, out-of-distribution property prediction, and molecular representation learning. The paper also includes a case study on antimicrobial peptide (AMP) design.

# Key strengths include:
- A simple but effective architectural unification of predictive and generative tasks.
- Broad evaluation across widely used benchmarks.
- Strong empirical performance, including improved OOD prediction.

# Key weaknesses include:
- Alternating attention is closely related to prior multitask transformer work (e.g., UniLM, Dong et al. 2019).
- Some dataset details are insufficiently documented or missing.
- Certain benchmark metrics (FCD, KL div., etc.) are used without definition, potentially confusing to readers outside the molecular ML community.

**Audience:**

Yes

**Audience Explanation:**

The work addresses a long-standing challenge in molecular ML which is jointly unifying generative and predictive modeling. Both the methodological contribution and the practical results are relevant for TMLR’s readership in machine learning and computational biology.

**Broader Impact Concerns:**

I do not see significant broader impact concerns beyond what the authors have already noted. The paper briefly acknowledges potential dual-use (e.g., generation of toxic molecules), which seems sufficient. The primary contributions are methodological and benchmark-driven. I do not think an expanded Broader Impact Statement is strictly necessary.

**Claims And Evidence:**

Yes

**Claims Explanation:**

The evidence is generally convincing, with experiments, baselines, and supplementary details. However, there are a few outstanding issues requiring clarification:

- Exact version or data of acquisition of the used resources for the GuacaMol and AMP datasets.
- Lacking information of the generation process of the AMPs ( at least number and size of generated sequence).
- Missing definition for the used metrics "Diversity" even in the referenced Li et al., 2024 paper.
- Explanation for the higher than expected ClinTox scores (99.5) in Table 3.

**Requested Changes:**

# Critical fixes:
- Dataset version or acquisition date (also state which AMP training examples are predicted only)
- Resolve issue with "Diversity" in section 5.3
- Provide justification for the higher than expected ClinTox scores (99.5) in Table 3.
- KL div. in an ML paper should be only used for Kullback–Leibler divergence (not a derived benchmark score)

# Benchmark metric definitions:
- Explicitly define or reference all benchmark metrics used (at least in the supplement). For example:
- FCD, KL div., Val., Uniq., Nov. in Table 4 (GuacaMol).
- Esol, Freesolv, Lipo, BBBP, BACE, ClinTox, Tox21, ToxCast, SIDER, HIV datasets in Tables 2, 3, 5.
- Even if these are standard in chemistry/ML, they should be clarified for a computer science audience.

# Novelty statement:
Clearly distinguish Hyformer’s alternating-attention mechanism from prior work such as Dong et al. 2019. A transparent discussion of similarities/differences will strengthen positioning.

# Minor changes:
- Fix the typo in Sec. 5.3: “Arginine (R) and Arginine (K)”  - (“Arginine (R) and Lysine (K)” ??)
- Ensure all acronyms are defined at first mention (e.g., FCD, SD, QED).
- Add explicit reporting of acceptance rates and predictor calibration for conditional sampling.

---

> ### Author Response · Authors · 2025-10-17
> **Updated Definitions to Reviewer fGqQ**
>
> # Updated Task Definitions
>
> ## Conditional Molecule Generation (Section 5.1.1)
>
> **Quantitative Estimate of Drug-likeness (QED).** A continuous metric of the drug-likeness of a molecule based on physicochemical properties such as molecular weight and hydrophobicity, with values ranging from 0 to 1. [1]
>
> **Synthetic Accessibility (SA).** A continuous metric quantifying how  difficult a molecule is to synthesize, derived from structural complexity, where lower values indicate easier synthesis. [2]
>
> **Partition Coefficient (logP).** A continuous metric of molecular hydrophobicity, defined as the logarithm of the partition coefficient between octanol and water, where higher values denote greater affinity for lipophilic environments. [3]
>
> Metric values calculated using rdkit 2023.09.2.
>
>
> ## Out-of-Distribution Molecular Property Prediction (Section 5.1.2)
>
> **DRD2-Hi.** Binary classification dataset of 8482 compounds with labels indicating dopamine receptor inhibition, with therapeutic relevance in schizophrenia and Parkinson’s disease; dataset obtained from ChEMBL30 [4].
>
> **HIV-Hi.** Binary classification dataset of 41127 compounds from the Drug Therapeutics Program AIDS Antiviral Screen, with labels indicating the inhibition of HIV replication; dataset obtained from MoleculeNet [5].
>
> **KDR-Hi.** Binary classification dataset with labels indicating VEGFR2 (vascular endothelial growth factor receptor 2) inhibition, a kinase target in cancer therapy, with training restricted to 500 compounds to simulate low-data regimes; dataset obtained from Chembl30 [4].
>
> **Sol-Hi.** Binary classification dataset of 2173 compounds with labels indicating solubility; dataset obtained at Biogen [6] .
>
> For further dataset and train/test splitting details, see [7]. Data accessed from <https://github.com/SteshinSS/lohi_neurips2023/tree/main/data/hi> [accessed 20.03.2023].
>
>
> ## Molecular Representation Learning (Section 5.1.3) and Molecular Property Prediction (Section 5.2.2).
>
> **ESOL.** Regression dataset containing water solubility measurements for 1128 compounds.
>
> **FreeSolv.** Regression dataset containing experimentally measured hydration free energy values in water for 642 compounds.
>
> **Lipophilicity.** Regression dataset containing experimentally measured octanol/water distribution coefficients (logD at pH 7.4), a key indicator of membrane permeability and solubility, for 4,200 compounds.
>
> **BACE.** Binary classification dataset of 1513 compounds with experimentally determined qualitative binding results for a set of inhibitors of human β-secretase 1 (BACE-1).
>
> **BBBP.** Binary classification dataset of 2039 compounds with binary labels indicating blood–brain barrier permeability.
>
> **ClinTox.** Multitask classification dataset of 1478 compounds with labels indicating whether a compound is (i) FDA-approved and/or (ii) failed clinical trials due to toxicity reasons.
>
> **HIV.** Binary classification dataset of 41127 compounds from the Drug Therapeutics Program AIDS Antiviral Screen, measuring inhibition of HIV replication.
>
> **Tox21.** Multitask classification dataset of 7831 compounds with qualitative toxicity measurements across 12 biological targets, including nuclear receptors and stress response pathways.
>
> **ToxCast.** Multitask classification dataset of 8575 compounds with qualitative toxicity results across over 600 in vitro assays, derived from high-throughput screening.
>
> **SIDER.** Multitask classification dataset of 1427 approved drugs, with side effects grouped into 27 system organ classes according to MedDRA classifications, capturing adverse drug reactions across organ systems.
>
> For further details, see [Table 1 in 5]. To ensure comparability with Uni-Mol [8], we accessed data from <https://bioos-hermite-beijing.tos-cn-beijing.volces.com/unimol_data/finetune/molecular_property_prediction.tar.gz> [accessed 20.03.2023].

---

> ### Author Response · Authors · 2025-10-17
> **Updated Definitions (cont.)**
>
> # Updated Metric Definitions
>
> **MAD.** Mean Absolute Deviation between predicted and target property values; lower is better.
>
> **SD.** Standard Deviation of generated property values from the target; lower is better.
>
> **Validity.** Fraction of syntactically valid molecules generated by the model; higher is better.
>
> **Uniqueness.** Fraction of unique molecules among generated samples; higher is better.
>
> **Novelty.** Fraction of generated molecules not present in the training set; higher is better.
>
> **KL Div. Score.** Score based on the Kullback–Leibler Divergence between various descriptor distributions of generated and training molecules; values normalized in the range [0, 1]; higher values indicate a closer match between descriptor distributions between generated and training molecules. For details, see [9].
>
> **FCD Score.** Score based on the Fréchet ChemNet Distance between the generated and reference (training) molecule embedding distributions, calculated in ChemNet feature space; values normalized in the range [0, 1]; higher values indicate closer resemblance of the generated to reference molecules. For details, see [9].
>
> **Perplexity.** Exponentiated negative log-likelihood of a sequence, with the log-likelihood being calculated per token, using ProGen2-medium [10]; lower values indicate greater model-based plausibility of the generated peptides.
>
> **Diversity.** Average pairwise Levenshtein distance between the generated sequences; higher values indicate greater diversity of the generated samples. For details, see Eq. 6 in [11], where Hyformer replaces Soergel with Levenshtein distance.
>
> **Fitness.** A measure quantifying to what extent a peptide forms a stable, amphipathic α-helix, computed according [12].
>
> **HydrAMP MIC.** The probability of a peptide being active against E.Coli bacteria strain predicted with HydrAMP [13].
>
> **AMPlify.** The probability of a peptide being antimicrobial predicted with AMPlify [14].
>
> **amPEPy.** The probability of a peptide being antimicrobial predicted with amPEPy [15].
>
>
> # References
> [1] Bickerton, G. R., et al. Quantifying the chemical beauty of drugs. Nature Chemistry. 2012.
>
> [2] Ertl, P. & Schuffenhauer, A. Estimation of synthetic accessibility score of drug-like molecules based on molecular complexity and fragment contributions. Journal of Cheminformatics. 2009.
>
> [3] Wildman, S. A., & Crippen, G. M. Prediction of physicochemical parameters by atomic contributions. Journal of Chemical Information and Computer Sciences. 1999.
>
> [4] Mendez, D., et al. Chembl: towards direct deposition of bioassay data. Nucleic acids research. 2019.
>
> [5] Wu, Z., et al. Moleculenet: a benchmark for molecular machine learning. Chemical science. 2018.
>
> [6] Fang C., et al. Prospective validation of machine learning algorithms for absorption, distribution, metabolism, and excretion prediction: An industrial perspective. Journal of Chemical Information and Modeling. 2023.
>
> [7] Steshin, S. Lo-hi: Practical ml drug discovery benchmark. NeurIPS. 2023.
>
> [8] Zhou G., et al. Uni-mol: A universal 3d molecular representation learning framework. ICLR. 2023.
>
> [9] Brown, N., et al. GuacaMol: benchmarking models for de novo molecular design. Journal of chemical information and modeling. 2019.
>
> [10] Torres, M. D. T., et al. Generative latent diffusion language modeling yields anti-infective synthetic peptides. 2025.
>
> [11] Jintae K., et al. Comprehensive survey of recent drug discovery using deep learning. International Journal of Molecular Sciences. 2021.
>
> [12] Li, T., et al. A Foundation Model Identifies Broad-Spectrum Antimicrobial Peptides against Drug-Resistant Bacterial Infection. Nat Commun. 2024.
>
> [13] Szymczak P., et al. Discovering highly potent antimicrobial peptides with deep generative model hydramp. 2023.
>
> [14] Li C., et al. AMPlify: attentive deep learning model for discovery of novel antimicrobial peptides effective against WHO priority
> pathogens. 2022.
>
> [15] Lawrence T. J., et al. amPEPpy 1.0: a portable and accurate antimicrobial peptide prediction tool. 2021.

---

> > ### Author Response · Authors · 2025-10-17
> > **Response to Reviewer fGqQ**
> >
> > We thank the reviewer for the thorough and constructive feedback.
> >
> > **Dataset version or acquisition date.**
> >
> > Guacamol. Data accessed from https://figshare.com/projects/GuacaMol/56639 on 20.03.2025
> >
> > AMP. Data accessed from Peptipedia <https://app.peptipedia.cl/>, Uniprot <https://www.uniprot.org/uniprotkb?query=%28length%3A%5B8+TO+50%5D%29>, AMPSphere <https://ampsphere.big-data-biology.org/downloads> and HydrAMP <https://drive.google.com/drive/folders/1krim1ugqNDmgmHZCFSOvmynWxCSzyOto> on 17.04.2025 with train/test set constructed using standard scikit’s `train_test_split` using `random_state=44`.
> >
> >
> > **Resolve issue with "Diversity" in section 5.3.** We added the following definition of Diversity in the Appendix, together with other metric definitions and stating the correct reference.
> >
> > **Diversity.** Average pairwise Levenshtein distance between the generated sequences; higher values indicate greater diversity of the generated samples.For details, see Eq. 6 in [11], where Hyformer replaces Soergel with Levenshtein distance.
> >
> > [11] Jintae K., et al. Comprehensive survey of recent drug discovery using deep learning. International Journal of Molecular Sciences. 2021.
> >
> > **Justification for the higher than expected ClinTox scores (99.5) in Table 3.** We hypothesize that a high ClinTox score may be due to a high imbalance of the ClinTox dataset, with positive to negative label ratio of 15.44 and 0.07, for each of the targets respetively. To test this hypothesis, we additionally report the F1 score for Hyformer, per target label. We find that the F1 score across labels for Hyformer is equal to 0.98 and 0.9, furhter highlighting Hyformer's strong perfomance on the linear probing experiment.
> >
> > **KL div. in an ML paper should be only used for Kullback–Leibler divergence (not a derived benchmark score).**
> > In the final version of the manuscripts, we will correct “KL divergence” to “KL divergence score”. Additionally, we will correct "Frechet ChemNet Distance" (FCD) to "Frechet ChemNet Distance Score" (FCD Score).
> >
> > **Lacking information of the generation process of the AMPs ( at least number and size of generated sequence).** For the antimicrobial peptide design experiment (Table 6 in Section 5.3), we undonditionally sample ~6.7M sequences and accept 50K of them, which results in an acceptance rate of ~1%. Please note that our sampling procedure intentionally prioritizes high selectivity over throughput. All peptides have a maximum lenght of 50 AAs.
> >
> > **Benchmark metric definitions.** We add an explicit list of task and metric definitions used throughout the paper. See Official Comment: Updated Definitions and Updated Definitions (cont.)
> >
> > **Novelty statement.** We thank the reviewer for this comment. Our novelty is two-fold. First, we apply an alternating self-attention scheme during both pre-training and fine-tuning, yielding a joint model that unifies generation and prediction. In contrast, Dong et al. (2019) employ alternating attention only during pre-training. Second, upon training we combine LM-based objectives (LM + MLM for pre-training and LM for fine-tuning) with a predictive task (Eq. 8). In contrast, Dong et al. (2019) employ only LM-based objectives: left-to-right LM, right-to-left LM, bidirectional LM (MLM) and sequence-to-sequence language modeling tasks, without any predictive task.
> >
> > **Typos and acronyms.** We thank the reviewer for pointing out these details. We have corrected the typo in Section 5.3, changing “Arginine (R) and Arginine (K)” to “Arginine (R) and Lysine (K)”. All acronyms (e.g., FCD, SD) will be now defined upon first mention, in the final version of the manuscript. We additionally revised the manuscript for further typos.
> >
> > **Acceptance rates and predictor calibration for conditional sampling.** We add a table indicating the number of accepted samples per 100K sampled examples in conditional molecule generation experiment with mean and std. dev. across 3 random seeds. The table confirms that sampling with Hyformer is highly selective. Additionally, in the final version of the manuscript we will add scatterplots of predicted vs. ground truth values for all three targets: QED, SA and logP for this experiment.
> >
> > Additional Table: Number of accepted samples per 100,000 generated molecules. Mean and std across three random seeds. Average (Avg.) calculated over all target values.
> >
> > | Hyformer   | QED=0.5 | QED=0.7 | QED=0.9 | SA=0.7 | SA=0.8 | SA=0.9 | logP=0.0 | logP=2.0 | logP=4.0 | Avg. |
> > |:----------|:--------:|:--------:|:--------:|:--------:|:--------:|:--------:|:----------:|:----------:|:----------:|:------:|
> > | **no_joint** | 315 (7) | 367 (11) | 162 (11) | 144 (5) | 448 (15) | 229 (23) | 14 (3) | 76 (7) | 70 (2) | 203 (81) |
> > | **joint**    | 295 (19) | 330 (4) | 176 (17) | 140 (10) | 426 (13) | 254 (7) | 11 (4) | 70 (7) | 67 (4) | 197 (71) |
> >
> >
> > We thank the reviewer again for the detailed suggestions, which will help further strengthen clarity and reproducibility.

---

### Review · Reviewer_ookq · 2025-10-08

**Summary Of Contributions:**

A great contribution unifying generative and predictive models for molecules.

The authors provide a great solution for SMILES-based generation, which does not yet exist in the literature.

They propose and alternating mask for the same transformer model, which is a clever and simple approach to achieve a seamless joint training. The authors give mathematical explanations of why there can be no conflicts, typical for joint learning.

The model excels or is on par to representation learning tasks, conditional and unconditional generation, as well as property predictions.

Well done!

**Audience:**

Yes

**Audience Explanation:**

Very appropriate contribution on generative modeling and property prediction of molecules. I think the most interesting aspect is that for realistic scenarios (e.g. OOD testing and/ or AMP design), it works well.

**Claims And Evidence:**

Yes

**Claims Explanation:**

Very clear presentation overall.

**Requested Changes:**

What does validity measure in Table 1? Could you clarify in the text?

What are the confidence intervals in Table 1?

Clean up the "the the" in the text.

---

> ### Author Response · Authors · 2025-10-16
> **Response to Reviewer ookq**
>
> We sincerely thank the Reviewer for their positive evaluation of our work.
>
> **Validity (Table 1).** Validity measures the fraction of syntactically valid SMILES strings generated by the model. This definition has been added to the text. We will additionally, as requested by Reviewer xgqr, include an explicit definition of all tasks and metrics used throughout the paper in the final version of the manuscript.
>
> **Confidence intervals (Table 1).** We added std. dev. calculated over 3 random seeds for Hyformer results in Table 1, as likewise requested by Reviewer xgqr. Similiarly, we additionally update Table 8 that contains a detailed breakdown of the conditional generative performance.
>
> Table 1: Conditional generative performance on GuacaMol dataset. Mean and std. dev. over 3 random seeds. Average (Avg.) is computed across all targets. Best model is marked **bold**.
>
> | Model     | Joint | Metric             | QED             | SA              | logP            | Avg.            |
> |------------|:------:|--------------------|------------------|------------------|------------------|------------------|
> | **Hyformer** | ✗ | MAD ↓        | 0.031 (0.003)   | 0.015 (0.001)   | 0.131 (0.010)   | 0.059 (0.004)   |
> |            |   | SD ↓          | 0.045 (0.004)   | 0.020 (0.001)   | 0.170 (0.014)   | 0.078 (0.006)   |
> |            |   | Validity ↑    | 0.993 (0.003)   | 0.990 (0.004)   | 0.985 (0.014)   | 0.989 (0.007)   |
> |            | ✅ | MAD ↓        | **0.008 (0.001)** | **0.005 (0.000)** | **0.041 (0.002)** | **0.018 (0.001)** |
> |            |   | SD ↓          | **0.015 (0.002)** | **0.009 (0.002)** | **0.051 (0.004)** | **0.025 (0.003)** |
> |            |   | Validity ↑    | 0.990 (0.007)   | 0.985 (0.003)   | 0.987 (0.006) | 0.987 (0.005)   |
>
>
> Table 8: Conditional generative performance on GuacaMol dataset across all targets Mean and std. dev. over 3 random seeds. Average (Avg.) is computed across all targets. Best model is marked  **bold**.
>
> | Model | Joint | Metric | QED=0.5 | QED=0.7 | QED=0.9 | SA=0.7 | SA=0.8 | SA=0.9 | logP=0.0 | logP=2.0 | logP=4.0 | Avg. |
> |:------|:------:|:--------|:--------:|:--------:|:--------:|:--------:|:--------:|:--------:|:---------:|:---------:|:---------:|:------:|
> | **Hyformer** | ✗ | MAD ↓ | 0.035 (0.000) | 0.032 (0.001) | 0.027 (0.007) | 0.020 (0.001) | 0.016 (0.000) | 0.009 (0.001) | 0.131 (0.012) | 0.135 (0.007) | 0.127 (0.011) | 0.059 (0.004) |
> |  |  | SD ↓ | 0.049 (0.000) | 0.046 (0.002) | 0.039 (0.010) | 0.027 (0.002) | 0.021 (0.000) | 0.012 (0.001) | 0.162 (0.015) | 0.174 (0.010) | 0.175 (0.016) | 0.078 (0.006) |
> |  |  | Validity ↑ | 0.993 (0.003) | 0.993 (0.003) | 0.993 (0.004) | 0.986 (0.009) | 0.985 (0.001) | **0.999 (0.002)** | 0.978 (0.031) | 0.983 (0.004) | **0.995 (0.007)** | 0.989 (0.007) |
> |  | ✅ | MAD ↓ | **0.010 (0.001)** | **0.009 (0.000)** | **0.006 (0.001)** | 0.008 (0.001) | **0.005 (0.000)** | **0.001 (0.000)** | 0.033 (0.005) | **0.044 (0.001)** | **0.046 (0.001)** | **0.018 (0.001)** |
> |  |  | SD ↓ | **0.018 (0.002)** | **0.018 (0.002)** | **0.010 (0.003)** | **0.015 (0.003)** | **0.009 (0.002)** | **0.004 (0.000)** | **0.037 (0.007)** | **0.057 (0.003)** | **0.059 (0.001)** | **0.025 (0.003)** |
> |  |  | Validity ↑ | 0.983 (0.009) | 0.990 (0.006) | 0.996 (0.005) | 0.976 (0.005) | 0.981 (0.002) | **0.999 (0.002)** | **1.000 (0.000)** | 0.991 (0.007) | 0.971 (0.011) | 0.987 (0.005) |
>
> **Typographical correction.** We have corrected the “the the” repetition noted by the reviewer. We have scanned the text for other typos.
>
> We thank the reviewer again for their encouraging feedback, which helped us further improve the manuscript.

---

### Decision · Action_Editor_uTrY · 2025-11-27

**Recommendation:** Accept with minor revision

**Additional Comments:**

I ask for minor revision as I see some inconsistencies in the boldings in multiple tables. Just as an example if we look at Table 2: Using three repeats, there is no significance test that would discriminate for example 0.784±0.082 from 0.782±0.062. Please revise and bold equivalent high performers in all cases. There is no basis claiming that one of the model is better than the other in these scenarios.

**Audience:**

Yes

**Audience Explanation:**

Reviewers unanimously agree that TMLR's audience would be interested in knowing the findings of this paper.

**Claims And Evidence:**

Yes

**Claims Explanation:**

Reviewers unanimously agree that the claims supported by convincing evidence.